# GII.17 norovirus re-emerged in the 2020s as a result of dynamic and adaptive evolutionary processes

Kentaro Tohma [1] ✉, Sonja Jacobsen[2], Britta Altmann[2], Joseph A. Kendra [1], Michael Landivar[1], William E. De La O[1], Maria Dolores Fernandez-Garcia [3,4], Karina A. Gomes [5], Sophia Chudnovsky[1], Lauren A. Ford-Siltz[1], Kelsey A. Pilewski[1], Yamei Gao[1], Ilya Mazo[6], Sandra Niendorf[2] & Gabriel I. Parra [1] ✉

Over the past two years, increased norovirus activity has been reported in multiple countries, accompanied by a rise in genotype GII.17 prevalence. Despite causing large outbreaks in Asia during 2014-2016, GII.17 has not historically been considered a predominant genotype. In this study, using 1471 archival and newly-identified GII.17 genomes, we investigated (i) global diversification patterns of this virus at the whole-genome level, (ii) in-depth mutational patterns within 511 viruses detected during a 10-year national survey in Germany, and (iii) intra-host viral diversity and adaptation processes that lead to the predominance of the GII.17 virus. The recent GII.17 norovirus exhibited extensive genetic diversity and multiple back-and-forth and recurrent mutations during the early phase of its epidemic; however, this diversity declined by 2024, suggesting that the virus had reached a phenotype efficient for human infection. Experimental data confirmed that mutations in the viral capsid enhanced binding to host factors associated with virus entry and resulted in antigenic changes compared to previously circulating clusters. Overall, this study demonstrated that the recent surge of GII.17 resulted from a dynamic, multifaceted process involving diverse adaptive strategies, ultimately enabling the virus to achieve sustained transmission within the human population.

Human norovirus is a major cause of acute gastroenteritis at all ages. Although norovirus disease self-resolves with 2–3 days in healthy individuals, it can lead to severe acute gastroenteritis in vulnerable populations[1]. It has been estimated that norovirus causes thousands of deaths per year globally, with most of them reported in middle- or low-income countries. In high-income countries, norovirus infections can result in significant economic losses[2]. Thus, norovirus vaccines are expected to have a significant benefit to global health[3].

The human norovirus genome consists of a single-stranded positive-sense RNA molecule that is organized into three open reading frames (ORFs): ORF1 encodes six nonstructural proteins (NS1/2-7) that are required for virus replication; ORF2, the major viral capsid

[1]Division of Viral Products, Center for Biologics Evaluation and Research, Food and Drug Administration, Silver Spring, MD, USA. [2]Department of Infectious Diseases, Robert Koch Institute, Berlin, Germany. [3]Enterovirus and Viral Gastroenteritis Unit, National Centre for Microbiology, Instituto de Salud Carlos III, Madrid, Spain. [4]CIBER Epidemiology and Public Health (CIBERESP), Madrid, Spain. [5]Laboratory of Viral Gastroenteritis, INEI-ANLIS "Dr. Carlos G. Malbrán", Buenos Aires, Argentina. [6]FDA HIVE, Center for Biologics Evaluation and Research, Food and Drug Administration, Silver Spring, MD, USA. ✉e-mail: kentaro.tohma@fda.hhs.gov; gabriel.parra@fda.hhs.gov

protein (VP1); and ORF3, the minor capsid protein (VP2). Because protective antibody responses to human norovirus are primarily directed against VP1, most vaccine strategies are based on this protein[3]. However, the extreme genetic diversity presented by VP1 has been one of the roadblocks for the development of efficacious norovirus vaccines. VP1 is structurally organized into the shell (S) domain, which forms the scaffold of the icosahedral particle, and the protruding (P) domain[4], which extends from the S and is further divided into conserved (P1) and variable (P2) subdomains. The latter contains the major B-cell epitopes involved in viral neutralization[5–8], as well as the binding sites for histo-blood group antigen (HBGA) carbohydrates[9]. HBGAs are molecules expressed on the surface of enteric epithelial cells that serve as attachment factors and facilitate norovirus infection[10–12]. Antibodies that block the interaction between HBGAs and VP1 have been shown to neutralize the virus[6,8,13,14]. Thus, in the absence of a robust cell system to cultivate and amplify human noroviruses, the HBGA-blockade assay has provided a reliable surrogate for evaluating viral neutralizing activity of antibodies.

Norovirus are classified based on genetic differences within ORF1 and ORF2, leading to a dual system for nomenclature. For example, GII.17[P17] viruses are classified within capsid type genogroup II genotype 17, and polymerase type GII.P17. Because the boundary of ORF1/ORF2 is a hotspot for recombination, different genotype and polymerase types can be described for a given virus, e.g. GII.17[P3], GII.2[P16][15]. Historically, GII.4 viruses are responsible for approximately 50% of infections among nearly 40 different human norovirus genotypes[16]. However, uncommon genotypes can occasionally rise to prominence in specific geographic regions or during certain seasons. One notable example was the predominance of GII.17 noroviruses in several Asian countries between 2014 and 2016[17]. Prior to this emergence, only a handful of GII.17 cases had been reported[18] (Fig. 1a). The increase of GII.17 cases during 2014-2016 was associated with the emergence of two clusters (namely, C and D[18,19]), which were genetically different from previously circulating clusters (namely, A and B) (Fig. 1b). The emergence of cluster C, first reported in Kawasaki, Japan[20], preceded the emergence of cluster D, which likely diverged from cluster C by rapidly acquiring several mutations on VP1[17,21]. As noted previously, viruses from clusters C and D displayed a new polymerase type, GII.P17, setting them apart from clusters A and B, which were associated with GII.P13, GII.P16, and GII.P31 polymerase types[18–20]. Cluster D viruses predominated in Asia and presented different antigenicity and a better affinity to different HBGA carbohydrates as compared to previously reported GII.17 viruses[22–25]. However, after this sudden increase, the number of GII.17 cases declined substantially by 2017, and GII.4 resumed its predominance worldwide[16,18]. Despite the emergence and predominance of clusters C and D, clusters A and B viruses are still reported circulating at low levels in different countries[18].

Over the past two years, increased activity of norovirus has been reported in multiple countries, accompanied by a rise in the number of GII.17 cases (Fig. 1a). This increase in GII.17 cases was first reported in Romania during a large outbreak of gastroenteritis in 2021[26]. Similar viruses were subsequently detected in Russia as early as 2021, where they became predominant during the 2022–2023 season[27]. These cases were associated with the emergence of a "new" GII.17 lineage that was surprisingly closer to cluster C viruses and divergent to the epidemic cluster D (Fig. 1b). Although several mutations have been identified that distinguish these new GII.17 viruses from cluster C viruses[26,27], their phenotypic characteristics remains unknown. Here, we conducted a comprehensive analysis of 1471 archival and recent GII.17 genome sequences at global, local, and intra-host levels to investigate how the virus has diverged and adapted to the human population over the past decade. Further, we experimentally demonstrated that a dynamic and adaptive evolutionary process, involving changes in antigenicity and binding to cellular attachment factors, contributed to the recent surge of this new lineage.

## Results

### Substitutions, insertions, and deletions play a role in shaping the diversity of GII.17 norovirus

To better understand the mechanisms leading to the re-emergence of GII.17 viruses during the 2020 s, we sought to determine how inter- and intra-cluster diversification contributed to the evolution of GII.17 norovirus. First, we regenerated the phylogenetic tree of GII.17 noroviruses using 1013 archival and recent GII.17 VP1 genome sequences, including 68 new sequences from viruses circulating in Germany, Spain, and Argentina (Fig. 1b; Supplementary Data 1 and 2). The phylogenetic analysis confirmed that the GII.17 viruses detected during 2021–2024 originated from cluster C viruses, diverging in parallel with cluster D viruses, and are associated with the GII.P17 polymerase type[26–28]. A closer look at the phylogenetic tree reveals that the new GII.17 viruses diverged into four sub-clusters: (i) ancestral viruses detected during 2014-2016, (ii) a sub-cluster of viruses first detected in Romania and Russia in 2021[26,27] and subsequently detected in other countries during 2021–2024, (iii) a sub-cluster of viruses only detected in the United States and Canada during 2022–2024, and (iv) those responsible for most outbreaks in Europe and the Americas during 2023–2024[28,29] (Fig. 1b). Interestingly, clusters C and D largely predominated in Eastern Asia during 2013–2016, while the new viruses are currently reported to cause large outbreaks in Europe and the Americas.

Next, we analyzed the differences of VP1 amino acid sequences among different clusters. No major inconsistencies were observed in phylogenetic clustering between trees generated using nucleotide and amino acid sequences (Supplementary Fig. 1). Clusters A and B presented ~40 amino acid differences among them, while clusters C and D were positioned distant from clusters A and B but relatively close to each other, with 20-30 amino acid differences between them (Fig. 1c). As expected from the phylogenetic tree, the viruses from the new cluster presented smaller differences from those in cluster C, differing by only 10–15 amino acids[27] (Fig. 1c). The new GII.17 noroviruses (2021–2024) acquired ~10 amino acid substitutions as compared to their ancestral virus E11161/France/2014. The Shannon entropy analysis suggested variable sites located differently in each cluster, therefore suggesting distinct evolutionary forces driving these changes (Supplementary Fig. 2). Although viruses from clusters C, D, and new viruses exhibited up to 10 intra-cluster amino acid changes, they displayed low entropy, indicating overall limited sequence variations within each cluster. As previously indicated, GII.17 noroviruses exhibited different lengths of their VP1 sequences due to several insertions and deletions (indels)[17,19,24] (Fig. 1d). Notably, this pattern is in stark contrast with the top three most predominant norovirus genotypes (GII.2, GII.3, and GII.4), which exhibit a limited number of indels during their genetic diversification. The indels on GII.17 norovirus clusters and two intermediate strains (Tokyo27-3/Japan/1976[30] and Arg13099/Argentina/2015[31], Supplementary Fig. 1) mapped at different loops of the outermost P2 subdomain of VP1 (Fig. 1e; Supplementary Data 3), resulting in unique patterns within or nearby the loops that interact with HBGA carbohydrates[9,24,32,33], and are analogous to the immunodominant antigenic sites from GII.4 noroviruses (Supplementary Fig. 3). The new GII.17 noroviruses showed the same indel patterns exhibited by cluster C[27] (Fig. 1e), supporting their evolutionary origin (Fig. 1b). Furthermore, we identified additional indels within the clusters that mapped to the T loop (residues 393-398). Thus, one cluster D virus presented an additional glycine (G) insertion, while others presented a G or aspartic acid (D) deletion, i.e., reverted to the cluster C pattern (Fig. 1f). Moreover, two new GII.17 noroviruses presented an insertion on the same T-loop. Taken together, these data demonstrate that GII.17

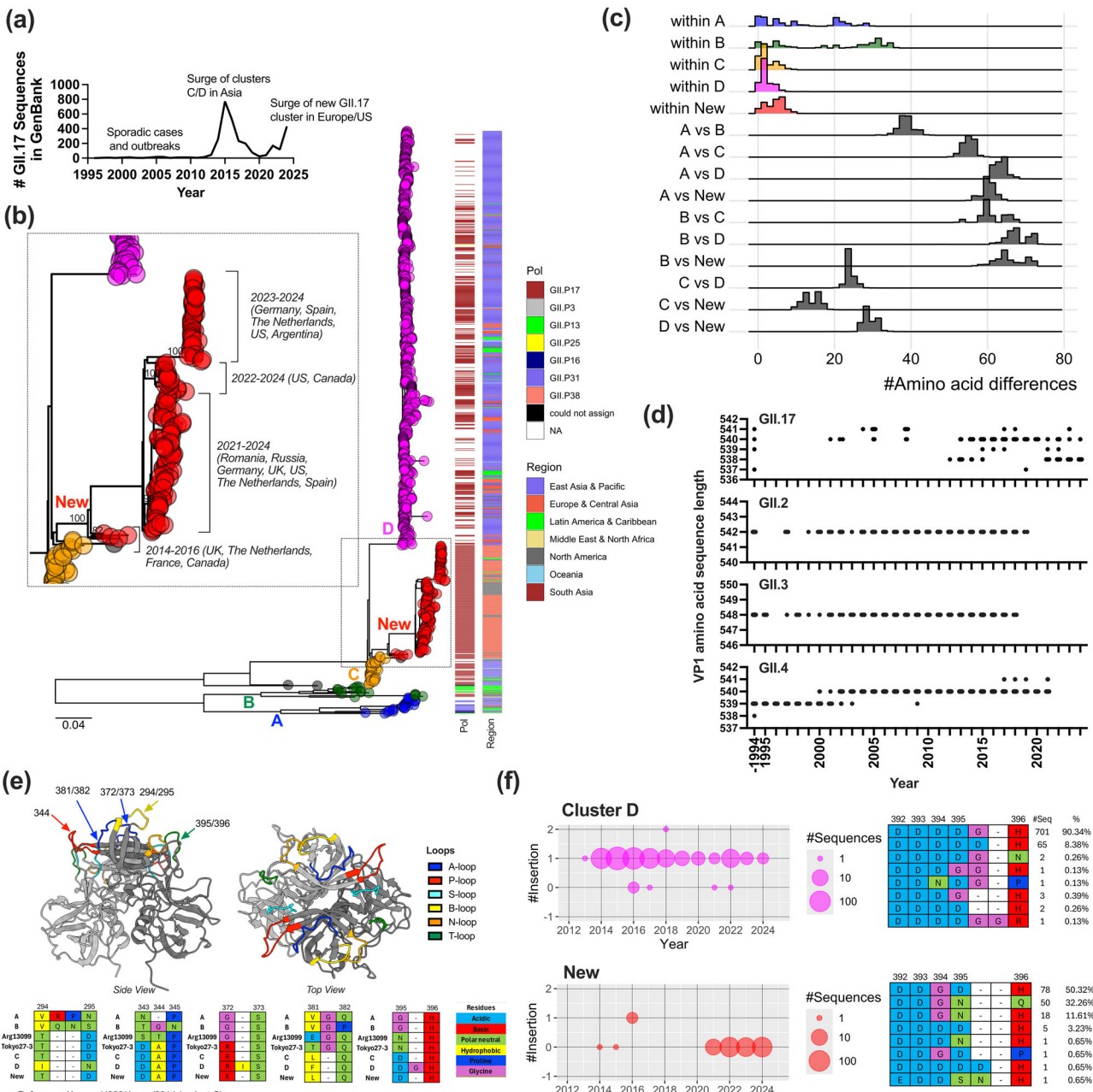

**Fig. 1 | Substitutions, insertions, and deletions on the VP1 shape the diversity of GII.17 noroviruses. a** The line graph summarizes the number of GII.17 sequences (≥100 nt in length) deposited in the GenBank since 1995 until 2024. **b** The maximum-likelihood tree indicates genetic diversity of 1,013 GII.17 noroviruses. Nearly complete VP1-encoding nucleotide sequences were used (n = 945 from GenBank; n = 68 from this study). Colors on the tips (indicated by circles) denote different GII.17 clusters, i.e., blue for cluster A, green for cluster B, orange for cluster C, magenta for cluster D, red for new GII.17 noroviruses, and gray for unique viruses. The heatmap on the side of the tree indicates corresponding polymerase types and geographic regions of the collection of the viruses. The inset subtree zooms in the viruses detected in recent outbreaks in Europe and America, with bootstrap values displayed on the nodes of the four clusters. **c** Pairwise VP1 amino acid differences were calculated and summarized at within- and between-clusters level in ridgeline plot. The y-axis indicates the density of each dataset.

**d** Temporal changes of VP1 sequence length of GII.17 and other dominant norovirus genotypes (GII.2, GII.3, and GII.4) was visualized using dot plots. The dataset for GII.2 (n = 801), GII.3 (n = 300), and GII.4 (n = 3,119) was retrieved from previous studies[86–88]. Sequences from animals, environment, and immunocompromised patients were excluded. **e** Structural positions and alignments of insertions and deletions of GII.17 noroviruses are rendered using structural data of the P domain (PDB#: 5F4M, Kawasaki323/Japan/2014). The alignments were colored by chemical properties of the amino acids. **f** Temporal distribution of intra-cluster indels detected in cluster D and new GII.17 noroviruses is summarized in bubble plot. The y axis indicates the number of insertions on position 395/396 as compared to Kawasaki323, cluster C virus. The positive shift indicates additional insertions and negative shift indicates deletions. The number (and %) of different sequence patterns and their alignment are summarized on the right. Source data is provided on figshare (https://doi.org/10.6084/m9.figshare.29421056).

diversification is defined by distinctive amino acid substitutions and indels on the VP1.

## Unique ORF1 and ORF3 genes contribute little to viral replication

We next analyzed the genetic diversity of the new GII.17 viruses at ORF1 and ORF3 regions. Despite phylogenetic divergence, Chhabra et al. classified the polymerase of the new GII.17 cluster as still belonging to the GII.P17 polymerase type[28]. Analysis of the ORF1 sequences encoding the viral polymerase (NS7) from GII.P17 and other closely related polymerase types confirmed that multiple clusters of GII.P17 are associated with the different VP1 clusters (Fig. 2a). The GII.P17 from clusters C and D grouped together, while the one from new lineage located between the original GII.P17 and GII.P3, which was associated with viruses from cluster B and another genotype, GII.3. Similar to the VP1 clustering, the GII.P17 from new viruses showed divergence from their ancestral sub-cluster (2014-2016) to the recent sub-cluster (2023–2024). Interestingly, viruses from cluster B virus detected in 2023–2024 acquired the GII.P17 polymerase[28], which presented mutations that made them closer to the GII.P3 polymerase. This pattern was confirmed with trees generated using different proteins encompassing the entire ORF1 region (Supplementary Fig. 4). The intermediate nature of the polymerase types found in the new GII.P17 cluster do not appear to be the result of recombination (Fig. 2b). Rather, convergent evolutionary events may have resulted in the diversification of these clusters. In this regard, within the new GII.P17 ORF1 genes, we noted several recurrent and back-and-forth mutations (Fig. 2c; Supplementary Fig. 5). The same or similar cluster-defining mutations were detected from different common ancestors on the tree, suggesting that the new GII.17 viruses continuously explored different mutations to identify optimal residues to establish sustained transmission in the human population. It has been previously suggested that the acquisition of a small number of mutations in the polymerase could have a great impact on viral replication and viral load[34,35]; thus, we explored whether any of these mutations (or combinations) resulted in differences in the viral load among patients infected with the different GII.17 noroviruses. To this end, we used viral genome titer information recorded from patients (n = 669) infected with GII.17 in Germany. No significant changes in viral load were detected at any given year (Fig. 2d), and no significant differences (p = 0.333 in t test, two-sided) were detected between patients infected with cluster D or new virus (Fig. 2e).

The ORF3 sequences encoding VP2 from the different GII.17 viruses were also divided into distinct clusters (Fig. 2f). Unlike VP1, the VP2 of clusters C and D were tightly grouped together, and the new GII.17 viruses were relatively distant from them[27]. Interestingly, VP2 also presented multiple indels at the inter- and intra-cluster levels (Fig. 2g), with seven viruses from the new lineage presenting up to 3 deletions on their VP2. Since there is limited information on the function of VP2, the role of these mutations remains unknown.

## Adaptive mutations result in differential binding to various host cell attachment factors

Given the genetic differences presented by the GII.17 clusters, particularly within the P2 subdomain of the VP1, we developed virus-like particles (VLPs) from two to three representative viruses from each cluster and two intermediate strains: Tokyo27-3/Japan/1976[30] and Arg13099/Argentina/2015[31] (Supplementary Fig. 1; Supplementary Table 1). A total of 16 VLPs were successfully developed (Supplementary Fig. 6) and tested for their ability to bind carbohydrates expressed in the saliva from nine individuals (Fig. 3). Human saliva contains HBGA carbohydrates secreted from epithelial cells. The amount and profiles of carbohydrate secretion varies by individuals with different levels of fucosyl transferases (FUT2 and FUT3) activities[36]. As such, a variety of carbohydrate types were secreted in saliva samples tested in this

study, including Lewis[a], Lewis[b], Lewis[x], Lewis[y], H type-1, and H type-2 antigens with four major patterns (Fig. 3a). GII.17 VLPs exhibited clear differences in their binding patterns to HBGAs (Fig. 3b). Particularly, VLPs from clusters A and B bound only to saliva expressing H type-2 and Lewis[y] carbohydrates, while clusters C, D and new GII.17 bound to a broader range of saliva samples. Thus, as suggested in previous studies[23,25], the stronger and broader HBGA-binding observed in clusters emerging since 2013, including those from the new GII.17 lineage, may explain their emergence and high prevalence in the population.

A key residue, Y444 on the S-loop, was previously identified as a determinant of binding to A or B trisaccharide (derivative carbohydrates with N-acetyl-galactosamine or galactose on H antigens)[24,33]. Our genome database confirmed that all 971 viruses from cluster C, D, and the new cluster, but none from clusters A and B, presented the tyrosine (Y) at this position (Fig. 3c). Of note, Arg13099 virus possessed the Y444 residue, yet did not exhibit strong binding with saliva samples 5–9, demonstrating that other residues are also responsible for binding to the H antigens. Notably, viruses 446222/Romania/2021 and 144/Germany/2024 showed stronger binding to saliva samples 7-9, which present strong expression of the Lewis[a] antigen, as compared to the ancestral virus E11161/France/2014. A unique difference is detected between these three VLPs, in which the virus E11161/France/2014 has a lysine (K) at position 361 on the N-loop, while the other two have an arginine (R) (Supplementary Data 3). While this position was represented by R361 in the new viruses, it was dominated by glutamine (Q) in all other clusters (Fig. 3d). Temporal codon sequence analysis indicates that in 2014, ancestral viruses from the new lineage introduced a single substitution at the first position of the codon *CAA* (Q), resulting in *AAA* (K) (Fig. 3e). Two viruses from the ancestral cluster (2014-2018) occasionally exhibited an additional mutation at the second codon position, producing *AGA* (R); however, these viruses did not successfully spread in the population at that time. In 2021, viruses from this new lineage once again showed the codon *AGA* (R), which finally predominated in the population. A few viruses reverted to *AAA* (K) or showed synonymous change (*AGG*), but they belong to the minority. Interestingly, the K361R mutation seems to be the key driver to gain the ability to bind to HBGAs in saliva samples 8 and 9 (Fig. 3f). Site-directed mutagenesis K361Q or K361R in E11161/France/2014 VLPs demonstrated increased binding of these VLPs to saliva samples 8 and 9. Unexpectedly, the reverse mutation introduced in Kawasaki323/Japan/2013 and 144/Germany/2024 VLPs did not reduce their binding to these saliva samples; instead, they enhanced the binding of Kawasaki323/Japan/2013 to saliva sample 9. The 361 mutation locates outside the HBGA-binding sites (Fig. 3d), suggesting an indirect impact on the structural interaction between the viral capsid and HBGAs.

## VP1 mutations shape the antigenic diversity of GII.17 noroviruses

We then used the panel of VLPs to determine the antigenic relationships among the different viruses and clusters using HBGA-blockade assays (Fig. 4). Mouse hyperimmune sera raised against seven different VLPs, representing viruses from each cluster and three viruses from the new lineage, were tested for their ability to block the interactions between all VLPs (n = 16) and HBGA carbohydrates (saliva sample 1). The pairwise blockade EC50 titers are summarized in Fig. 4a, which were then transformed and projected to the 2D antigenic map using antigenic cartography[37] (Fig. 4b). The 2D antigenic map successfully captured the relationships demonstrated with the blockade EC50 titer table (Supplementary Fig. 7a). Overall, viruses from the same cluster were grouped together. As expected from the genetic differences (Fig. 1), viruses from clusters A and B, or clusters C, D and the new lineage, presented lower antigenic distances (Fig. 4b), as reflected by the weak to moderate levels of cross-reactivity detected among viruses from those clusters (Fig. 4a). The exceptions are Arg4446/Argentina/2005 and CA-RGDS-1180/US/2023 viruses, which clustered together

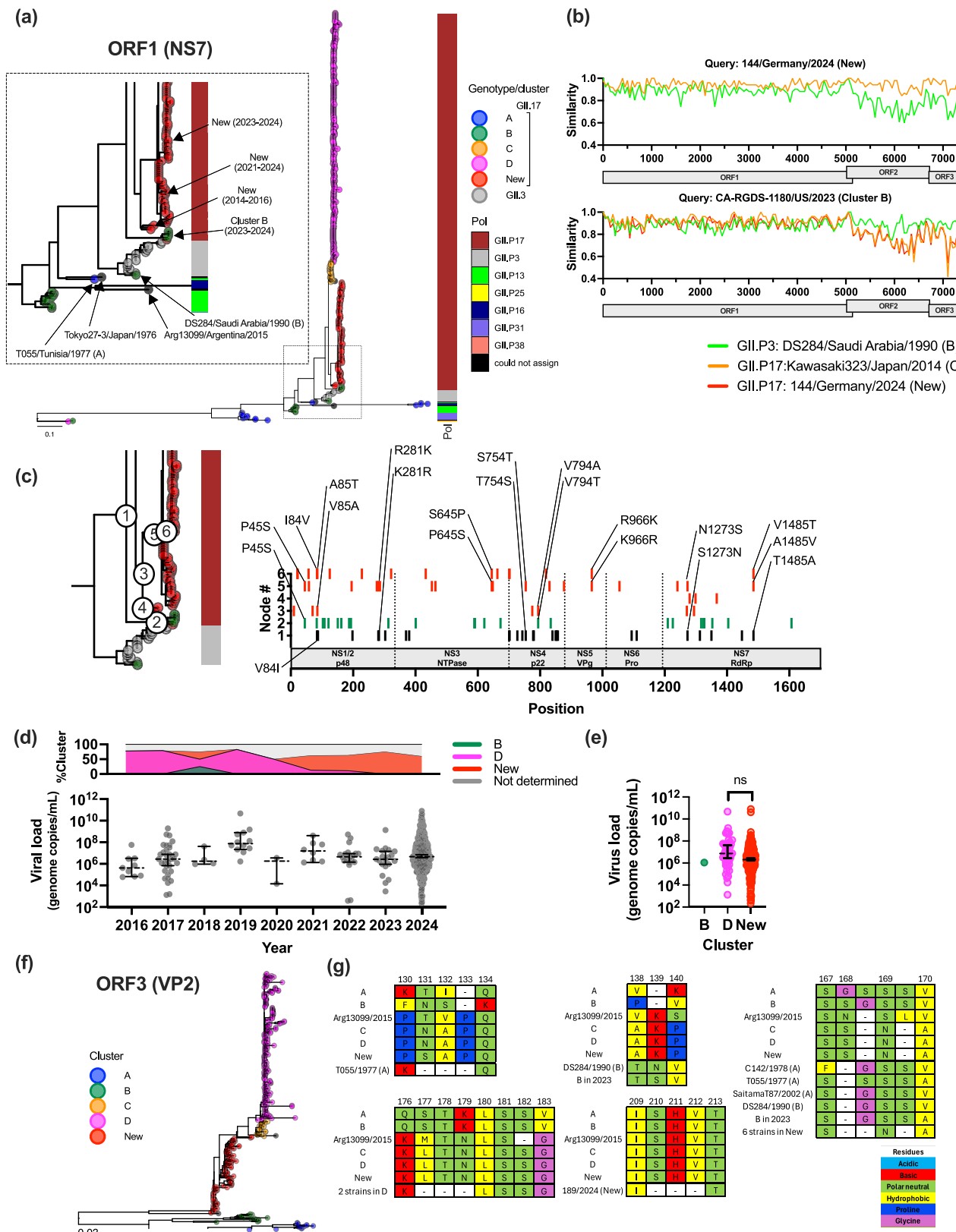

but located distant from other cluster A or B viruses. This is because they presented less blockade titers at both intra- and inter-cluster levels. Uncertainty in mapping of these two viruses are confirmed by bootstrap analysis (Supplementary Fig. 7). The oldest virus, Tokyo27-3/1976, was antigenically distant from the rest of the viruses, while the other unique strain, Arg13099/2015, was antigenically similar to clusters C, D, and the new viruses. Importantly, mouse sera against new

GII.17 viruses presented strong cross-reactivity to Kawasaki323, a cluster C virus, with lesser responses against cluster D viruses. Likewise, serum against cluster D virus did not cross-react well with the new viruses.

Analysis using antigenic distance (measured on the map) and genetic distance indicated strong correlation between them ($r = 0.813$), most of which was associated with differences on the

**Fig. 2 | Unique clusters of ORF1 and ORF3 in new GII.17 noroviruses with little impact on the viral load. a** Maximum-likelihood tree of 607 NS7-encoding nucleotide sequences indicates genetic diversity of GII.P17 and other closely related polymerase types. The tips on the tree (circles) were color coded by cluster (determined by VP1) following the same color scheme in Fig. 1. The heatmap next to the tree indicates corresponding polymerase types. The inset subtree zooms in the GII.P17 viruses detected from new GII.17 noroviruses and recent cluster B viruses. **b** Simplot-like line graphs show the genome-wide sequence similarity among different GII.17 viruses to identify potential recombination signals and break points. The colors of lines denote different viruses. **c** Ancestral reconstruction of amino acid sequences using 607 partial ORF1 sequences spanning NS1/2 to NS7-encoding regions indicates key cluster-defining amino acid mutations on the ORF1. In this panel, back-and-forth and recurrent mutations are indicated. A full list of mutations is described in Supplementary Fig. 5. **d** Viral load within 669 acute

gastroenteritis patients infected with GII.17 in Germany was measured using qPCR and summarized by years. Top areal plot shows the ratio of clusters detected between 2016 and 2024. The bottom dot plot shows qPCR titers (genome copies/mL) with median and the 95% confidence interval in each year. **e** The qPCR titers were compared between those infected with cluster B (n = 1), cluster D (n = 48), and new (n = 367) GII.17 noroviruses. The dot plot shows median and the 95% confidence interval of the qPCR titers. There was no significant difference detected between qPCR titers of cluster D and new viruses in t test (p = 0.333, two-sided). **f** The maximum-likelihood tree of VP2-enoding nucleotide sequences (n = 475) represents genetic diversity of VP2 among different GII.17 clusters. **g** The alignments show the insertion and deletions detected within VP2 amino acid sequences. The colors used within the alignment denote chemical properties of amino acids. Source data is provided on figshare (https://doi.org/10.6084/m9.figshare.29421056).

P domain (Fig. 4c). Correlation-scanning analysis, which calculated the correlation between genetic and antigenic distance using sliding windows of the genome region, identified the largest contribution to the overall antigenic diversity to the genetic differences on the P2 subdomain (Fig. 4d). Notably, the antigenicity-defining residues are located differently among clusters (Fig. 4d, middle and bottom panels).

Next, we aimed to identify the residues that contribute to antigenic differences among the various GII.17 clusters. Leveraging the extensive dataset generated from our antigenic analyses (Fig. 4a, b), we employed a machine learning approach. By pairing genetic information with antigenic distances for all viruses included in the study (Fig. 4; Supplementary Table 1), we used random forest regression analysis to predict which residues are most likely to define these antigenic differences. Residues (features) with the top 30 highest importance values were plotted on the P domain structure (Fig. 4e). When using the data from all pairs of viruses, 21 out of 30 residues mapped to the P domain, with 12 of them mapping to the loops. Notably, residues involving separation of C/D/new viruses clustered on the top center area of the VP1, while those separating all viruses (i.e., mostly A/B and C/D/new clusters) were scattered across the top surface of the VP1 (Fig. 4e; Supplementary Fig. 8). To validate our GII.17 analyses, we applied the same machine learning approach to antigenic cartography data from GII.4 noroviruses[38], for which the key residues underlying antigenic differences among variants have been experimentally defined[6,39–42]. Of the 34 residues known to define variable antigenic sites in GII.4, 17 were accurately predicted by our machine learning approach (Supplementary Fig. 9). These residues are among the most variable and functionally critical in distinguishing GII.4 variants[6,42,43], supporting the reliability of this approach in identifying residues that drive antigenic variability of GII.17 noroviruses.

Finally, we assessed the seroprevalence among 28 adult blood donors in the United States. Anti-GII.17 serum IgG titers remained low until the 2024–2025 winter season. However, serum samples collected in January 2025 –following the steep rise of GII.17 cases observed in December 2024– exhibited higher IgG titers against both the newly emergent GII.17 viruses and previously predominant cluster D viruses (Fig. 4f; Supplementary Fig. 10).

### Dynamic adaptive process during evolution within the local population and individuals

To gain further insight into the evolutionary dynamics leading to the emergence of the new GII.17 viruses, we conducted in-depth evolutionary analyses using data of 10 years of molecular surveillance of norovirus conducted by the national reference center in Germany (Fig. 5a). In Germany, cluster D viruses were detected from 2015 to 2022, until they were replaced by new viruses during the 2023–2024 season (Fig. 5b, c). The first detection of these new viruses occurred in 2018, when a virus similar to E11161/France/2014 (from the 2014 to 2016 cluster) was identified. Subsequent detections in 2021 revealed viruses resembling Romanian strains (2021–2024 sub-cluster)

and those from the 2022–2024 sub-cluster. After the sporadic detection of viruses from this lineage (n = 30, 2021–2024 sub-cluster; n = 3, 2022–2024 sub-cluster), epidemic strains (n = 356, 2023–2024 sub-cluster) finally emerged and spread in this country during the 2023–2024 season (Fig. 5b). Remarkably, phylogenetic trees using 511 P2 subdomain sequences collected during this surveillance period indicated quite different evolutionary patterns between cluster D and the new GII.17 noroviruses (Fig. 5b, c). The new viruses showed a sharp increase of effective population size (Ne) in 2024 while cluster D viruses presented gradual increase of Ne throughout the years. The new viruses exhibited higher entropy (amino acid variations) during the early phase of the epidemic, which declined in the later phase, whereas cluster D viruses maintained low entropy throughout the years. This temporal trend of variations was also confirmed using the global dataset (Supplementary Fig. 11). The new viruses exhibited more deleterious mutations (detected only on the tips of the tree) during the earlier phase of dissemination; meanwhile, cluster D viruses showed the majority of mutations on the nodes on the tree (i.e., mutations that persisted in the next generations). As a result, the new GII.17 viruses presented larger number of sites under diversifying selection as compared to cluster D viruses (Fig. 5b, c; Supplementary Data 4). Interestingly, many of these sites presented independent parallel (or recurrent, e.g., E293K, I355V, N374S/Y), as well as back-and-forth changes (e.g., D376N and N376D, D395N and N395D) over time (Fig. 5b; Supplementary Fig. 12), similar to what was observed in ORF1 (Fig. 2c). They were located on the top surface of the VP1 (Fig. 5d), overlapping with the potential (predicted) antigenic sites (Fig. 4e; Supplementary Fig. 3). Of note, K361R mutation, which increased HBGA-binding affinity of new GII.17 viruses in the 2020 s (Fig. 3), was not listed as a cluster-defining mutation. Only one virus belonging to the ancestral cluster was detected in Germany in 2018 and it occasionally presented R361. Two additional mutations (V334M and A350S), which define the later emerging clusters, were located at the VP1 dimer interface (Fig. 5d). Whether these mutations played an epistatic role in promoting the fixation of the K361R mutation or contributed to the structural fitness of the capsid in the dominant viruses remains unclear.

From the same dataset, we selected 49 individuals (4 individuals in 2021, 13 in 2022, 11 in 2023, and 21 in 2024) for intra-host mutational analysis of the new GII.17 noroviruses using next-generation sequencing (Fig. 5e; Supplementary Data 5). Interestingly, the average viral diversity within individuals also declined from the early to the late epidemic phases. Multiple cluster-defining and/or positively selected mutations (Fig. 5b, d) were detected within individuals as intra-host single nucleotide variants (iSNVs) (Fig. 5f). Except for one iSNV, M334L, viruses explored iSNVs which were always detected in the global population. The mutation R361K, which was associated with the HBGA-binding pattern, was also detected within multiple individuals but never took over the dominance of R361 at the population level. As demonstrated by the intra-host diversity at residue 361, multiple

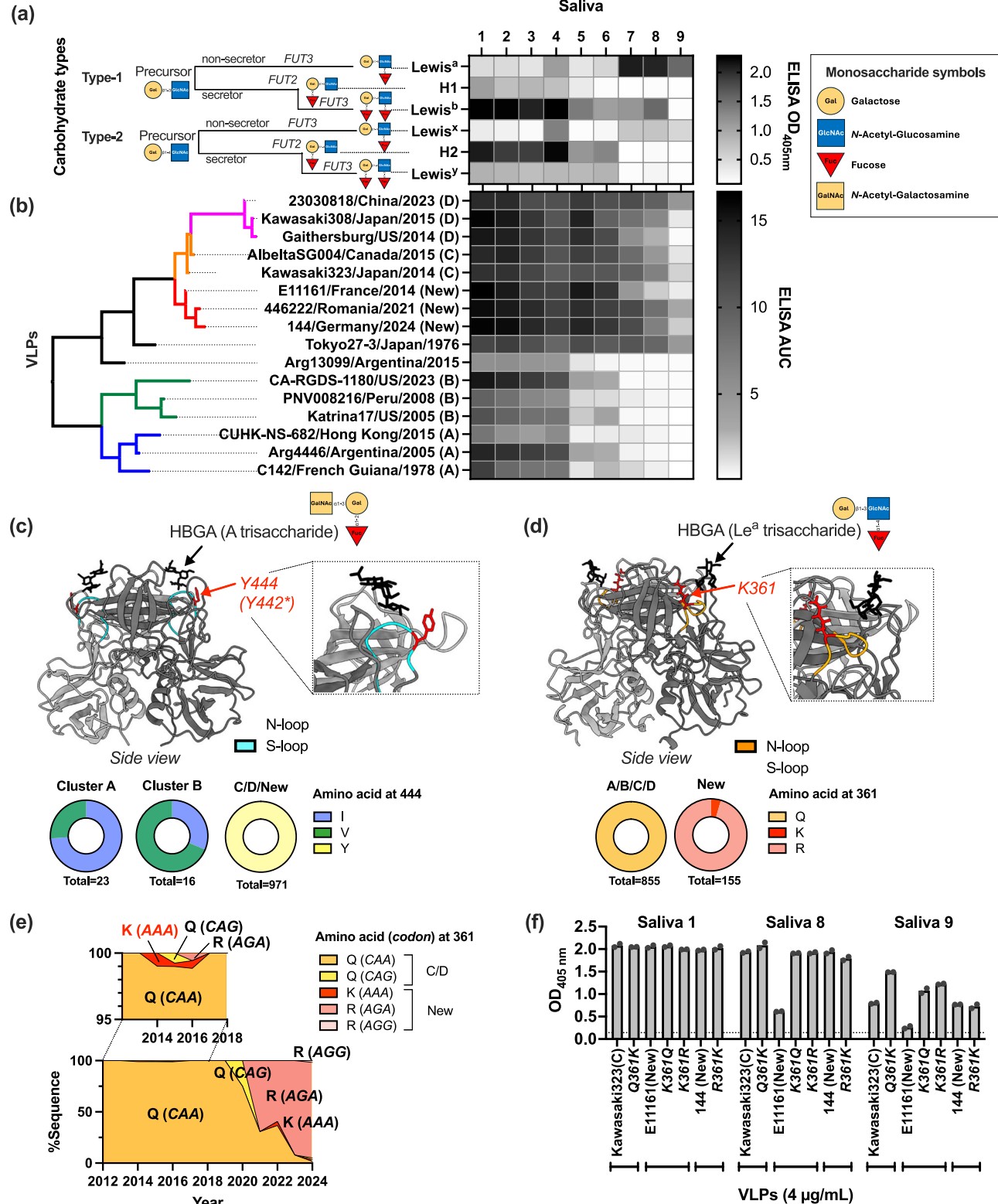

instances of iSNVs were detected across different individuals (Fig. 5g). However, we found no positive associations between the number of individuals sharing the same iSNVs and either their intra-host frequency or their prevalence in the local population. Thus, a high intra-host frequency of iSNVs does not appear to explain the number of individuals who presented the same iSNVs, and purifying selection simply eliminated the iSNVs detected in multiple individuals. Notably, those detected in multiple individuals, e.g., V363G, L366F, L366S,

mapped on the side surface of the P domain, while four iSNVs detected only in single individuals, F286L, S350A, P387L, and G389C, mapped internally in the VP1, suggesting structural constraints (Fig. 5g; Supplementary Fig. 3). In any case, they did not spread well into the population. We found that cluster-defining mutations and/or those under diversifying pressure detected within individuals presented higher intra-host frequency, and such iSNVs were often detected in the population (Fig. 5h), suggesting that adaptive selection played a role at

**Fig. 3 | New GII.17 noroviruses acquired a mutation to achieve better binding to additional HBGA cell-attachment factors.** Two heatmaps summarize (**a**) Histo-blood group antigen (HBGA) carbohydrate types secreted in different saliva samples as measured by ELISA OD$_{405nm}$ values of saliva (1:200 dilution) to different anti-HBGAs antibodies, and **b** their interactions to 16 different GII.17 VLPs presented by ELISA Area Under the Curve (AUC) titers using serial dilution of VLPs against different saliva samples. The left schematic in (a) illustrates biosynthesis process of different HBGAs. In both assays, mean of duplicates wells are presented. **c** Key determinant mutation on position 444 on cluster D (or *442 on Kawasa323) viruses to increase their binding to HBGAs[24,33]. All the 973 viruses from cluster C, D, and new viruses presented tyrosine (Y) at this position while viruses from clusters A and B presented isoleucine (I) or valine (V). Structural data of P domain and A tri-saccharide (mimics A antigens biosynthesized from H antigens) was rendered using PDB #5ZV5. **d** A single mutation K361R that differentiated ancestral and latest new GII.17 noroviruses, which presented different binding patterns to saliva samples 7-9

(non-secretors). All 855 sequences from other clusters showed glutamine (Q) at this position and the new viruses predominantly exhibited arginine (R), with lysine (K) in a minor population. Structural data of P domain and Le$^a$ trisaccharide was rendered using Alphafold-predicted structure of E11161/France/2014. **e** Codon and amino acid mutational pattern at position 361. Ancestral viruses acquired a single nucleotide mutation (*CAA* to *AAA*) to present Q361K mutation in 2010s, which remerged in 2020 s with an additional nucleotide change (*AAA* to *AGA*) to present K361R in 2020 s. **f** Mutagenesis assay confirmed that K361R (and K361Q) mutations on ancestral E11161/France/2014 virus gained binding to non-secretor saliva samples (8 and 9). The y axis shows ELISA OD$_{405nm}$ values using 4 µg/mL VLPs (a concentration that reached to saturation). The bar shows mean of duplicate wells and individual data points. The dashed horizontal line indicates OD$_{405nm}$ values from negative control. Source data is provided on figshare (https://doi.org/10.6084/m9.figshare.29421056).

## Discussion

The emergence of viruses is driven by the generation of genetic variations within hosts and their subsequent transmission in the population[44]. However, the factors determining the epidemic potential of novel viruses can vary. Here, we analyzed genetic and phenotypic data of an emerging lineage of GII.17 noroviruses that has been circulating at a low level for over a decade and recently caused large outbreaks in Europe and the Americas[28]. The data demonstrate that (i) viral loads in individuals infected with the new GII.17 lineage remained within the range observed for cluster D viruses despite the presence of mutations in nonstructural proteins (Fig. 2), but (ii) ancestral viruses from the new lineage acquired a single mutation in the viral capsid protein that improved their affinity to broader host cell-attachment factors (Fig. 3), (iii) additional mutations on the VP1 resulted in the evasion of immunity developed against the previously circulating predominant viruses (Fig. 4), and (iv) a dynamic adaptation process of this virus during the early epidemic phase resulted in a steady viral population during the later parts of the epidemic phase (Fig. 5). These events resemble those from the predominant norovirus genotype, GII.4, or other prominent human RNA viruses, e.g. influenza and SARS-CoV-2[45–47]. In contrast, they differ from those seen in other non-GII.4 noroviruses such as GII.2. The rise of GII.2 norovirus during 2016-2017 was not linked to changes in HBGA-binding profiles or antigenic differences but instead linked to recombination at ORF1/2 boundary and mutations in nonstructural proteins involved in replication[48]. This change was associated with higher viral load that was speculated to increase transmissibility[34], which was not the case of the recent GII.17 infections. Recombination did not seem to contribute to the emergence of the new GII.17 viruses. Instead, convergent evolutionary events, such as recurrent and back-and-forth mutations, shaped their ORF1, resulting in divergence from previously circulating GII.17 viruses. Notably, a recently detected cluster B virus also exhibited an ORF1 sequence that resembles the new GII.P17 without clear evidence of recombination, suggesting that these convergent changes may confer a fitness advantage. Further studies are warranted to determine whether mutations detected in nonstructural proteins play a role in other aspects of the viral cycle of the new GII.17 lineage[18].

Thanks to collaborative efforts aimed at enhancing genetic surveillance of noroviruses[49], we were able to conduct fine-scale analyses to uncover the evolutionary patterns of emerging noroviruses. Strikingly, GII.17 noroviruses are capable of successfully incorporating

in-frame indels into their genomes. Of the over 40 norovirus genotypes infecting humans, only a few (GI.3, GII.4, and GII.6) are known to introduce indels in their viral capsid protein that are maintained in the human population[50–52]. A well-known example of an insertion event occurred during the emergence of the GII.4 Farmington Hills variant[51]. This insertion has been associated with changes in HBGA-binding, antigenicity, and a change in the evolutionary process of GII.4 that resulted in the emergence of several variants during 2002-2012[6,42,51,53]. Interestingly, multiple in-frame indels have been detected in viruses infecting immunocompromised patients; however, these viruses have not successfully spread in the global population[39,54]. This suggests that while such changes are structurally tolerated, they may not consistently confer a selective advantage for widespread transmission. In GII.17, one insertion, within residues 395-396, was predicted to be associated with antigenicity (Fig. 4e). Although the machine learning model was not capable of predicting the impact of the insertion itself, it was successful in predicting the role of the substitution among cluster D viruses. In-depth genetic analysis revealed further indel events occurring at the intra-cluster levels (Fig. 1). Additionally, the new GII.17 noroviruses exhibited multiple recurrent and back-and-forth substitutions under diversifying pressure, affecting both the structural and nonstructural proteins. Contrasting evolutionary patterns –smaller diversification in cluster D versus greater variation in the new lineage viruses early in their emergence– suggest that immune pressure did not play a major role in such changes. The minimal variation observed in cluster D viruses may indicate that they achieved an optimally adapted phenotype after the quick transition from cluster C viruses[21,55]. In contrast, the variability observed in the newly emerged GII.17 noroviruses likely represents a transitional phase during which the virus is still evolving toward an optimally adapted phenotype for efficient transmission. This process seems to be driven by their fitness landscape involved in structural, replication kinetics, or host interaction factors, but not immune pressure. The ancestral virus was already antigenically different yet not completely adapted. Together, these data demonstrate the unique flexibility of the GII.17 norovirus in introducing mutations, enabling it to explore alternative changes that may alter its phenotype and enhance its fitness to infect the population. A better understanding on the structural and biological differences of the capsid of different norovirus genotypes can shed lights on how structural flexibility could benefit viruses to reach epidemic potential.

The rise of GII.17 viruses between 2013 and 2015 coincided with shifts in host susceptibility. Thus, several studies have shown that cluster C and D viruses bind HBGAs more effectively than the cluster A and B viruses[23,25]. One of the adaptive mutations acquired by the new GII.17 noroviruses is K361R, which improved the ability to bind HBGA carbohydrates and partially explained why the ancestral viruses from that lineage did not spread in the population. Indeed, this mutation

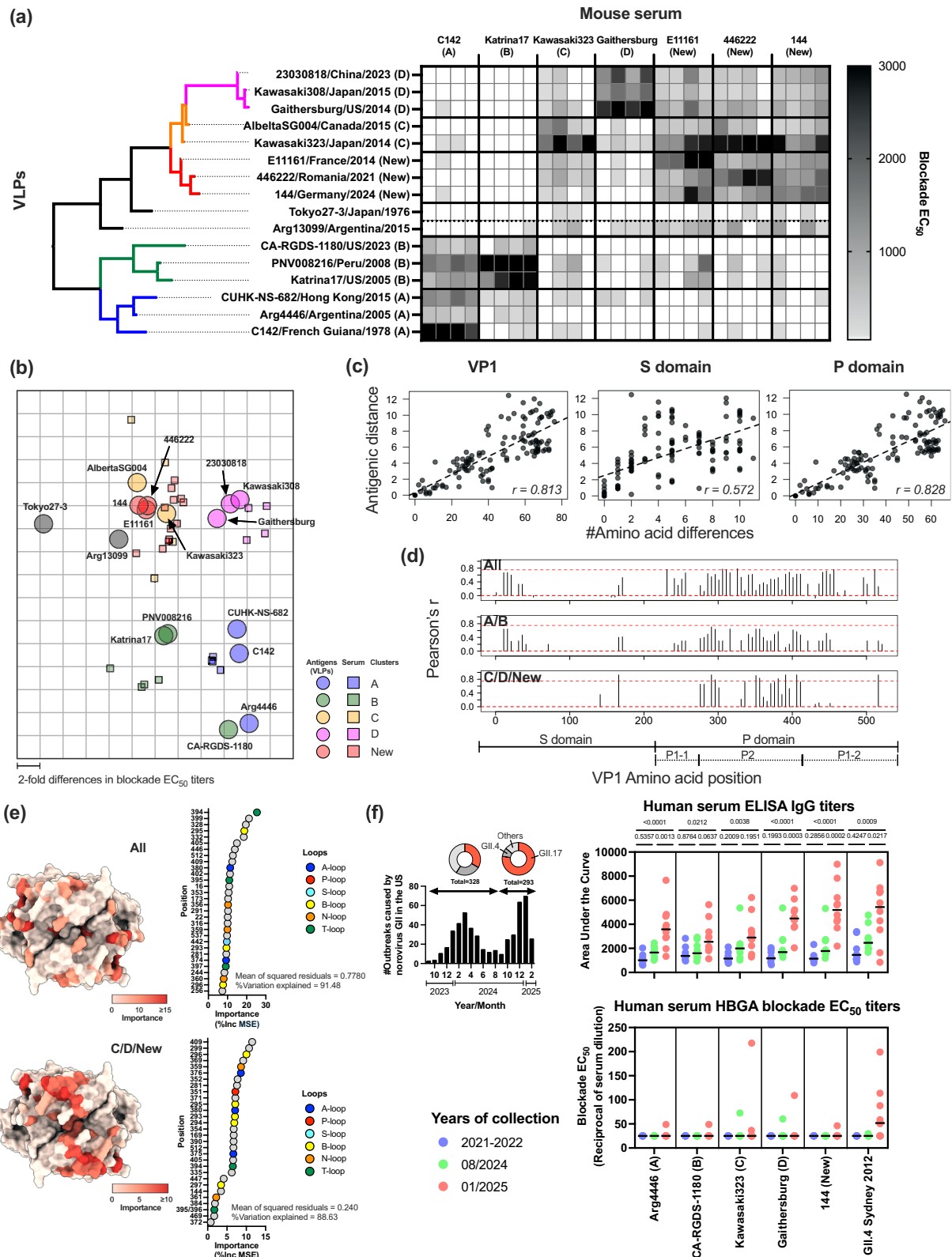

conferred the new viruses an enhanced ability to bind to the Lewis[a] and/or Lewis[x] antigens, which are expressed by non-secretor individuals. Previous studies have confirmed that HBGA profiles are associated with individual susceptibility to norovirus infection[12,56,57], and the previous epidemic caused by cluster D viruses was linked to their improved capacity to infect a broader range of host populations[23–25,58].

Thus, while the difference of binding patterns within the new viruses is small (adding minor non-secretors[59] in their susceptible population list) as compared with the large difference observed between clusters C/D and A/B viruses, the contribution of this single mutation to the new GII.17 noroviruses could have provided an evolutionary advantage over the ancestral strains of that lineage. Differences in non-secretor

**Fig. 4 | Mutations on the VP1 shape the antigenic diversity of GII.17 noroviruses.** **a** Histo-blood group antigen (HBGA)-blockade assays using 16 different VLPs and 7 groups (n = 4 each) of mouse serum immunized with different GII.17 clusters are summarized in the heatmap. Each cell indicates blockade titers from a pair of a serum and VLPs. **b** Two-dimensional antigenic map was created using HBGA-blockade titers. Larger circles represent antigens (VLPs) and smaller squares represent serum, color-coded by clusters. **c** The scatter plots show correlation between antigenic distance and genetic distance among pairs of VLPs tested; using entire VP1 (left), S (middle), and P domain (right) sequences. **d** The correlation scanning analysis indicates sites with highest correlation with antigenic distance. Top graph includes data using all pairs of viruses, the middle using clusters A and B, and the bottom from clusters C, D, and the new lineage. The y-axis shows Pearson's correlation coefficient (*r*) between antigenic distance and genetic distance within the sliding windows (window size = 50 amino acids, step size = 5 amino acids). The red dashed lines indicate *r* = 0 and 0.75. **e** Sites associated with antigenic differences were predicted using random forest method and mapped on the P domain structure (PDB#: 5F4M). The color gradient indicates importance of the positions (%Inc MSE; the increase of the mean squared error when given variable is randomly permuted) on the model, which are summarized in the dot plot at the right. The colors of the dot indicate HBGA-binding loops. **f** Seroprevalence of GII.17 noroviruses in the Unites States. The left bar graph summarizes the number of recent outbreaks caused by GII noroviruses in the United States. The right dot plots show IgG ELISA Area Under the Curve (AUC) and HBGA-blockade $EC_{50}$ titers of 28 human serum/plasma samples collected during pre- and post-epidemic periods of GII.17 noroviruses in the United States. The lines in the dot plots show the median values. Multiplicity adjusted p values in One-way ANOVA and Tukey multiple comparison test are provided for the ELISA data. Source data is provided on figshare (https://doi.org/10.6084/m9.figshare.29421056).

status have been described in different populations, e.g., weakened FUT2 activity in Asian population by missense mutation and completely inactivated FUT2 activity in Caucasian population by nonsense mutation[59]. Thus, these host and viral differences may, in part, explain the emergence of cluster C and D viruses in Asian countries during 2013-2015 and the new viruses in Europe and the Americas during 2023-2025.

Regarding the antigenic diversification of GII.17 noroviruses, several factors may have contributed to the recent emergence and predominance of new GII.17 cluster. These include reduced population exposure to noroviruses during the COVID-19 pandemic due to physical restrictions[60,61], overall low seroprevalence to GII.17 norovirus, anamnestic antibody responses due to previous infections with cluster D viruses, and limited cross-reactivity of cluster D-specific antibodies against new GII.17 viruses. In addition to providing a comprehensive antigenic characterization for this genotype, this study also offers novel insights into potential residues involved in the antigenic evolution of GII.17 noroviruses and a novel approach to identify residues involved in the antigenic diversification of human noroviruses. The machine learning approach implemented in this study successfully predicted the involvement of amino acid residues that have been experimentally validated for cluster D viruses, such as residues 293-299[62] and 393-396[22]. Moreover, the model also accurately predicted at least half the previously characterized antigenic sites of GII.4, further supporting the use of this computational framework to define residues that contribute to the antigenic diversity of noroviruses and other fast-evolving viruses.

Limitations of this study include (i) we were unable to experimentally confirm the phenotypic effects, such as differences in viral fitness, associated with the various mutations detected, particularly those located in the polymerase or VP2 of the new lineage, (ii) several of the residues predicted to be associated with antigenicity were not experimentally validated with site-directed mutagenesis, (iii) the genomic data used in our analysis are geographically limited, primarily originating from only a few countries, (iv) the viral loads presented need careful interpretation as they could be affected by sampling biases, such as timing of sample collection post-infection or host-related factors, and (v) it remains unclear why viruses similar to Tokyo27-3/Japan/1976 did not predominate in earlier years despite exhibiting antigenicity and HBGA-binding profiles comparable to those associated with the predominance of modern GII.17 clusters. Regardless of these limitations, this study demonstrates that the surge of a new GII.17 lineage was achieved via multiple different strategies throughout their evolution and provides critical data that can be used for the development of medical countermeasures against norovirus. Viruses from this cluster have been detected in Japan and other Asian countries over the past few months (e.g., GenBank accession: LC868426). Thus, it remains to be determined whether this virus will continue to disseminate and cause large outbreaks worldwide.

# Methods
## Viral genome sequencing
We retrospectively analyzed 531 GII.17 norovirus sequences collected as part of the routine surveillance conducted in Germany (n = 511), Spain (n = 16), and Argentina (n = 4) (Supplementary Data 1).

In Germany, fecal samples from norovirus positive cases (gastroenteritis outbreaks or sporadic cases) were sent to the Consultant Laboratory for Norovirus by local public health authorities, diagnostic laboratories, or physicians from all regions. For genotyping of norovirus circulating in the country, RT-qPCR was conducted for the first screening of norovirus and to determine the viral titers (viral genome copies/mL 10% stool suspension) of norovirus positive samples, followed by genotyping using conventional PCR and Sanger sequencing of partial ORF1 and ORF2 (P2 subdomain) sequences as described previously[63]. In addition to genotyping of capsid and polymerase types, 49 GII.17 norovirus samples collected from different locations (federal states) and spanning different years were subjected to whole-genome sequencing (Supplementary Data 5). To this end, double-stranded cDNA was synthesized either manually or using the Biomek i7 automated liquid handler (Beckman Coulter) as described previously[64]. Uniquely dual-indexed libraries were generated in an automated workflow using the NEBNext Ultra II FS DNA Library Preparation Kit (New England BioLabs Inc.) following the manufacturer's instructions on the Biomek i7. Target enrichment was performed by hybrid-capture using 714 of customized 80 nucleotides baits with flexible 4x tiling designed and synthesized by a service provider (Daicel Arbor Bioscience), followed by the myBaits hybridization capture for targeted NGS protocol (version 5.02). The customized nucleotides baits were designed using 470 reference norovirus sequences listed on Supplementary Data 6. The selection of these sequences was based on the computational tool developed by Metsky et al. [65]. Between 8 to 10 libraries with approximately equal viral load were pooled when available. The pools were then concentrated with a MinElute PCR Purification Kit (Qiagen) and subjected to capture hybridization for 24 h at 65 °C. Following the hybridization and washing steps, the enriched DNA was amplified using the KAPA HiFi Library Amplification Kit (KAPA Biosystems) for 14 cycles and purified using 1.8× MagSi-NGSPREP-Plus beads (Steinbrenner Laborsysteme). The final quantification was performed using Qubit dsDNA HS Assay kit on a Qubit Flex fluorometer (Life Technologies) and fragment size distribution was assessed using TapeStation 4150 (Agilent Technologies) with the High Sensitivity D1000 ScreenTape kit. The final pools were normalized to 2 nM and sequenced on the Illumina NextSeq 1000/2000 platform (Illumina Inc.) for 2 million reads of 2×150 bp. The generated fastq files were analyzed using FDA High-performance Integrated Virtual Environment (HIVE) platform[66]. The reference-based mapping, alignment and profiling were conducted using HIVE-hexagon Aligner v2.1[67] and HIVE-heptagon profiler v2.1[68] to generate consensus sequences.

In Spain, norovirus infections are not notifiable. However, routine genotyping is performed to characterize norovirus outbreaks reported

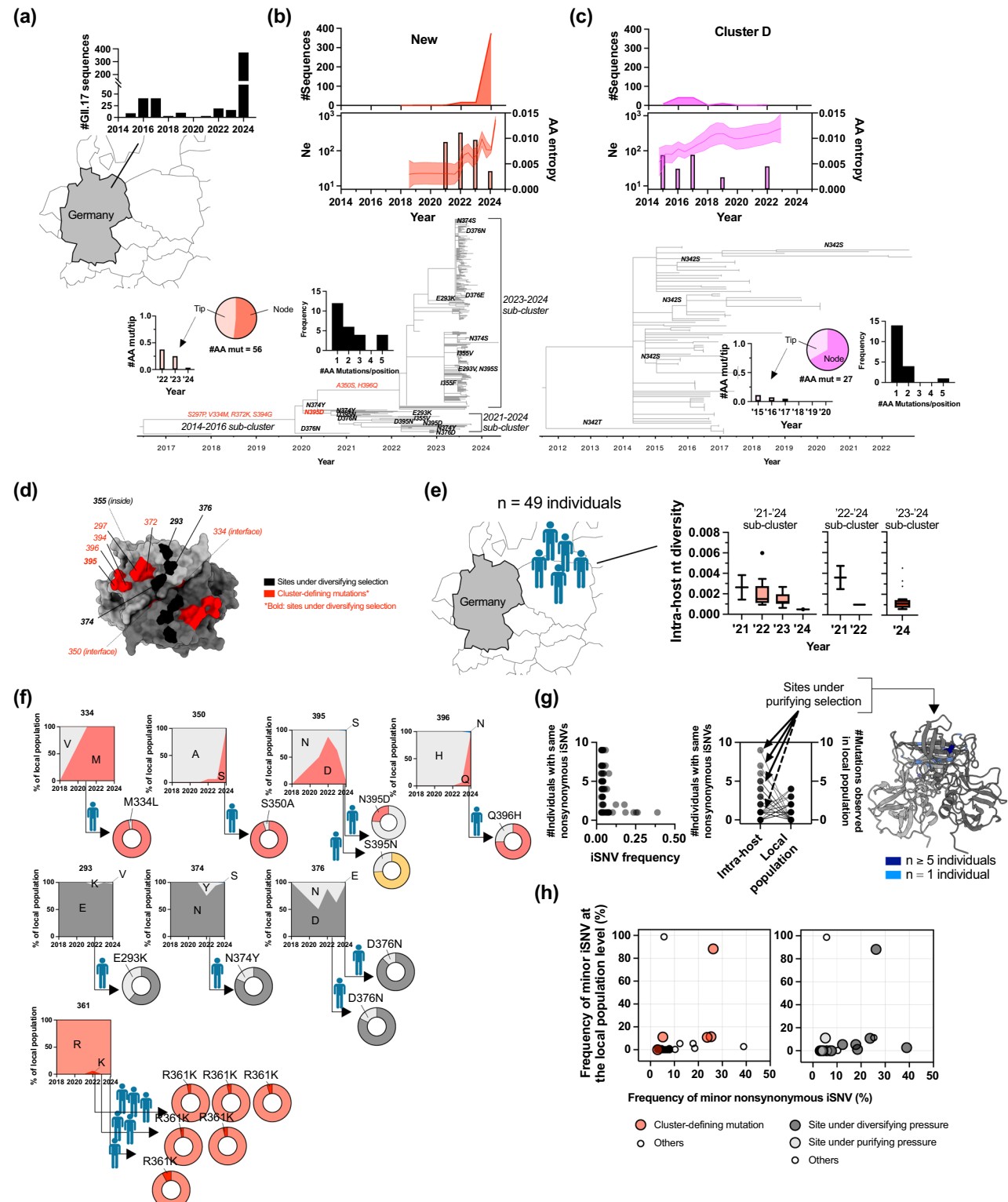

by autonomous communities to the National Center for Microbiology. Viral identification was performed using metagenomic next-generation sequencing approach as described previously[69], and we selected norovirus GII.17 positive cases for further analyses in this study. Briefly, RNA was extracted from 10% stool suspensions using the Quick RNA viral kit (Zymo). Samples were dual indexed during the library preparation conducted using the NEBNext Ultra II Directional RNA Library Prep Kit for Illumina with NEBNext Multiplex Oligos (New

England BioLabs Inc.). Target enrichment was performed by hybrid-capture using the Twist Comprehensive Viral Research Panel v2 (Twist Biosciences, catalog# TW-103550). Hybrid-capture libraries were combined equally by mass into capture pools (average 5-plex). Twist post-capture library pools were PCR amplified 16 cycles. Enriched libraries were sequenced using a Nextseq 500 Illumina platform (Illumina Inc) with a NextSeq 500/550 High Output Kit v2.5. SPAdes v3.14.0 in metaSPAdes mode was used to perform a de novo assembly of

**Fig. 5 | Dynamic adaptive process of new GII.17 noroviruses during their evolution within local population and individuals. a** GII.17 norovirus circulations since 2015 until 2024 in Germany. **b** P2 subdomain of 402 new lineage and **c** 107 cluster D viruses were subjected to evolutionary analyses, providing effective population size (Ne; middle line graphs, maximum-likelihood estimates and ±2×SD confidence bounds), temporal change of Shannon entropy (middle, bar graphs), time-calibrated phylogenetic trees, and mutational patterns (bottom, trees). The mutational patterns were classified into those detected on the tips or nodes (pie chart). Ratio of mutations on the tips was summarized by years (left bar plot of the pie chart). The frequency of mutations per position was summarized in the histogram right next to the pie chart. Cluster-defining mutations (red) and those under diversifying selection (bold) are indicated on trees. **d** Positions of key adaptive mutations are highlighted on the P domain structure (PDB#: 5F4M). **e** 49 individuals were assessed for intra-host viral diversity analyses. The Tukey's box plots (median and 25th–75th percentiles) summarize temporal trend of intra-host nucleotide

diversity by different genetic clusters. **f** Areal plots indicate temporal trend of major/minor mutations detected in Germany population (n = 402) with pie charts indicating those detected within individuals as intra-host single nucleotide variants (iSNVs). **g** Scatter plot indicates negative association between iSNV frequency and number of parallel events (#individuals with same iSNVs, n = 82 iSNVs), which do not explain the transmissibility of those to the populations (aligned dot plot, n = 69 mutations). Majority of mutations independently detected from different individuals were under purifying pressure at the population levels, with those detected from ≥5 individuals mapped on side surface of the P domain while those detected from only single individuals mostly mapped internal. **h** Scatter plots show associations between iSNV frequency and their prevalence in the local population (n = 82), colored by cluster-defining mutations (left) or types of selective pressures on positions of those iSNVs (right). The geographic map was rendered using sf v1.0-21 and spData v2.3.4 packages in R v4.5.0. Source data is provided on figshare (https://doi.org/10.6084/m9.figshare.29421056).

non-host reads[70]. Contigs taxonomic annotation was based on alignment to the NCBI's viral database using BLAST+ v2.9.0. A reference based-mapping approach was also used to obtain complete consensus viral genomes using Bowtie2 v.2.4.4[71].

In Argentina, three gastroenteritis outbreaks were reported across three provinces. Stool and vomit samples from each outbreak were analyzed, and norovirus GII.17 was identified as the causative agent in all cases. Nearly full-length sequences of GII.17 noroviruses were then obtained using amplicon-based next generation sequencing as part of a recently-published study[29].

All sequences were submitted to the GenBank as listed in Supplementary Data 1. The raw sequencing reads were deposited in Sequence Read Archive (SRA) (BioProject accession number: PRJNA1279050). SRA BioSample accession numbers are listed in Supplementary Data 5. Human-derived sequences were removed from next-generation sequencing files before uploading them to the SRA using NCBI human RefSeq genome assembly GRCh38.p14 (GCF_000001405.40) database and BWA v0.7.19-r1273[72], SAMtools v1.22[73], and Seqtk tool v1.5 (https://github.com/lh3/seqtk).

### Evolutionary analyses

Adding to the newly generated GII.17 sequences as described above, we collected all the publicly available GII.17 norovirus sequences (n = 945; ≥1500 nt in length) from GenBank on October 18, 2024. The criterion of ≥1500 nt in length was designed to include as many sequences as possible in order to capture a comprehensive view of the evolutionary history of this genotype without losing information from variable regions on the VP1. To analyze the divergence patterns of ORF1 sequences, we also retrieved sequences of non-GII.17 noroviruses with polymerase types genetically related to those of GII.17 from GenBank (n = 18). The GenBank accession numbers, sequence length, and corresponding metadata of retrieved sequences are summarized in Supplementary Data 2. The sequences were multiple-aligned using MAFFT v7 (online version)[74] and separated into different ORFs using MEGA v11[75]. Final datasets consist of 1,471 GII.17 and 18 non-GII.17 sequences, split into 416 NS1/2, 417 NS3, 421 NS4, 425 NS5, 456 NS6, 607 NS7, 1,013 VP1, and 475 VP2-encoding sequences including 68 newly obtained sequences in each dataset. In addition, we created P2 subdomain sequence dataset, which includes 511 sequences from viruses collected in Germany (1 each from cluster B and C, 107 from cluster D, 402 from new cluster). The capsid genotypes and polymerase types were confirmed using Norovirus Typing Tool v2.0[76]. Phylogenetic trees were inferred using maximum-likelihood method and best-fit substitution model as implemented in IQ-TREE multicore v1.6.12[77], in which ambiguous bases (e.g., gaps, missing positions) were ignored/nulled during the estimation of genetic relationship. Branch support was evaluated using 1000 runs of ultrafast bootstrap analysis. Time-calibrated trees (uncorrelated relaxed clock), effective population size (skyline plot), ancestral sequences of the nodes, and

mutational patterns were estimated using TreeTime v0.11.4[78]. The trees were visualized using R v4.5.0 and ggtree package v3.16.0[79] or FigTree v1.4.4 (https://github.com/rambaut/figtree/releases). Amino acid differences among viruses were calculated using phangorn package v2.12.1 in R v4.5.0[80]. Normalized Shannon entropy values of amino acid sequences on P2 subdomain were calculated using HDMD package v1.2 in R. The recombination analysis was conducted using R. The genetic similarity within 200 nt in 50 nt-shifted sliding windows were calculated using phangorn package v2.12.1 and visualized as Simplot-like plot. The diversifying and purifying selection were predicted using FEL and FUBAR (grid size = 50) methods as implemented in Datamonkey Adaptive Evolution Server[81]. Cutoff of posterior probability ≥0.9 in FUBAR and p ≤ 0.1 in FEL was used to determine sites under diversifying/purifying selection. The structural data (PDB #5F4M for Kawasaki323/Japan/2014 from cluster C virus and PDB #5ZV5 for 41621/China/2014 from cluster D virus) was visualized using USCF ChimeraX v1.9[82]. The structure of E11161/France/2014 was predicted using AlphaFold v2[83] implemented in the FDA HIVE platform, which provided good alignment score of RMSD = 0.99 as measured using Pairwise Structure Alignment tool in RCSB PDB.

### Intra-host viral population analyses

The iSNVs from 49 individuals with GII.17 infections in Germany (Supplementary Data 5) were detected using HIVE-heptagon profiler v2.1[68] with depth of coverage cutoff of ≥10 and quality score ≥20. The iSNVs detected in each run were summarized and annotated to identify the position, coding protein, and mutation effect (e.g., nonsynonymous, synonymous, indels) using in-house Python script. The iSNV profiles were visualized using graphics package v4.5.1 in R v4.5.0 or GraphPad Prism v10. The intra-host viral diversity was measured using pairwise nucleotide diversity ($\pi$) as described previously[44] using R v4.5.0. Statistical tests were conducted using R v4.5.0 or GraphPad Prism v10.

Before proceeding to detailed analyses of iSNVs, the quality and sensitivity analyses were conducted to determine the cutoff filters of iSNV calling (Supplementary Fig. 14). To identify potential sequencing errors, three samples were selected to repeat the entire process of RNA extraction and next-generation sequencing. As compared to amplicon-based deep sequencing approach[84], metagenomic-based sequencing generated less errors. The average number of iSNVs and ratio of synonymous and nonsynonymous iSNVs were similar regardless of the filters of duplicate or not (Supplementary Fig. 14a-b). The higher iSNV frequency cutoff reduced the number of iSNVs from >4000 at 0.1% cutoff to <1000 iSNVs at 1% cutoff and <200 at 3% cutoff, while depth of coverage did not affect the detection rate of iSNVs (Supplementary Fig. 14c). Sensitivity analysis using two different frequency cutoff values (3% and 5%) confirmed that the overall data were very similar in both sensitive and conservative filters; thus, intra-host nucleotide diversity at different genome positions were similar in both filters

(Supplementary Figs. 14d). The distribution of iSNV frequency were well correlated between the 3% and 5% frequency cutoff ($R^2 = 0.9598$, Supplementary Fig. 14e), and annual trend of intra-host nucleotide diversity was same (Supplementary Fig. 14f). Based on these data, we selected iSNV frequency of 3% and depth of coverage of 100 as cutoff values.

### GII.17 norovirus VLPs production

To characterize the HBGA-binding repertoire and antigenicity of GII.17 noroviruses, the VP1-encoding sequences of representative GII.17 viruses (Supplementary Table 1) and mutants (position 361 swapped) were synthesized and cloned into pFastBac1 vector (GenScript, catalog# SC1691). Using Bac-to-Bac Baculovirus Expression System (Gibco), the pFastBac1 vector was transformed into DH10Bac competent cells to produce a bacmid, which was transfected into Sf9 insect cells (ATCC) to express the VP1 protein. The expressed VP1 proteins were self-assembled to form VLPs and purified using 25% sucrose cushion (4 h, $98,400 \times g$ at $4\,^{\circ}C$) and cesium chloride gradient (18 h, $280,000 \times g$ at $15\,^{\circ}C$), followed by dialysis with 1×PBS (pH 7.4, Gibco) using Slide-A-Lyzer Dialysis Cassette (Thermo Scientific). Expression of VP1 protein was confirmed by western blot with norovirus VP1-specific mouse monoclonal antibody 30A11 (1:10,000 dilution)[6] and goat anti-mouse IgG secondary antibody conjugated with horseradish peroxidase (HRP) (SeraCare, Cat#5220-0339, 1:2000 dilution). VLP integrity was confirmed by transmission electron microscopy using uranyl acetate as staining solution (Supplementary Fig. 6). The concentration of the VLPs was measured using Qubit Protein Assay Kit (Invitrogen).

### HBGA carbohydrate characterization in saliva

Human saliva samples were purchased (Innovative Research) or collected (IRB protocol number: CBER IRB 16-069B and NIAID IRB 11-I-0109) from nine adult individuals. Saliva samples were boiled at $100\,^{\circ}C$ for 10 min and centrifuged at $15,871 \times g$ for 5 min to obtain saliva supernatant. To characterize HBGA carbohydrate types secreted in the saliva, saliva supernatant (1:200 in 1×PBS, pH 7.4) was coated on 96-well "U" bottom microtiter plates overnight at $4\,^{\circ}C$ and blocked for 1 h at room temperature with 5% blocking buffer (Bio-Rad). 1:100 dilution of anti-Lewis$^a$ (Sigma-Aldrich, Cat#SAB4700762, clone 7LE), Lewis$^b$ (abcam, Cat#AB3968, clone 2-25LE), Lewis$^x$ (Calbiochem, Cat# 434631, clone P12), Lewis$^y$ (Calbiochem, Cat#434636, clone F3), H type-1 (Invitrogen, Cat#14-9810-82, clone 17-206), and H type-2 (Invitrogen, Cat#MA1-35386, clone 19-OLE) mouse monoclonal antibodies were incubated on the saliva-coated plates for 2 h at room temperature. After three washes with 1×PBS, 0.1% Tween 20, plates were incubated with 1:2,000 dilution of goat anti-mouse IgA + IgG + IgM secondary antibody conjugated with HRP (SeraCare, Cat#5220-0342) for 1 h at room temperature and washed three times. Binding signals were visualized using ABTS 1-Component Microwell Peroxidase Substrate Kit (SeraCare). The $OD_{405nm}$ values were measured using SPECTROstar Nano plate reader (BMG LABTECH) and visualized as a heat map using GraphPad Prism v10.

### HBGA-binding assay

To test binding of VLPs to different carbohydrates, saliva supernatant (1:200 in 1×PBS, pH 7.4) was coated on 96-well "U" bottom microtiter plates (Costar) overnight at $4\,^{\circ}C$ and blocked for 1 h at room temperature with 5% blocking buffer (Bio-Rad). Fourfold serial dilution of 4 μg/ml wild-type or mutant VLPs were incubated on the saliva-coated plates for 1 h at $37\,^{\circ}C$. Plates were washed four times with 1×PBS, 0.1% Tween 20. 1:5,000 dilution of pooled sera from guinea pigs immunized with C142/French Guiana/1978 (cluster A), PNV008216/Peru/2008 (cluster B), Kawasaki323/Japan/2015 (cluster C), and Gaithersburg/US/2014 (cluster D) GII.17 VLPs were incubated for 1 h at $37\,^{\circ}C$ to detect the VLPs attached on the saliva-coated plates. After four washes, VLPs binding on the HBGA carbohydrates were incubated with 1:2,000

dilution of goat anti-guinea pig IgG-HRP (SeraCare, Cat#5220-0366) for 1 h at $37\,^{\circ}C$ and washed four times. Binding signals were visualized using ABTS and $OD_{405nm}$ values were measured using SPECTROstar Nano plate reader (BMG LABTECH). Measured values were summarized using AUC (Area under the curve) using GraphPad Prism v10.

### Animal immunization

Hyperimmune sera against wild-type GII.17 VLPs were produced in four female BALB/cAnNCr mice (Charles River Laboratories, strain code #555) with five to six weeks of age at the beginning of the immunization, following a schedule of intramuscular VLP immunization (40 μg VLPs/dose) with alum (Alhydrogel adjuvant 2%, CRODA) and two boosts on weeks 4 and 8 as described previously[42]. Cardiac terminal bleed was performed on week 10. Serum was separated from whole blood using BD Microtainer separator tubes (BD Diagnostics). Maximum five mice were housed in single cages with access to food and water, and maintained following FDA Veterinary Services guidelines under 12-h light/dark cycle at approximately $22\,^{\circ}C$ and 45% relative humidity. Animal protocol was approved by the FDA Institutional Animal Care and Use Committee (IACUC) (protocol number 2018-41). Hyperimmune guinea pig sera, used for the detection of norovirus VLPs in the HBGA-binding and blockade assays, were produced in previous study[85] under the approved animal protocol (FDA IACUC protocol number 2017–29).

### HBGA-blockade assay

To determine the serum antibody titers against different GII.17 VLPs, the HBGA-blockade assays were conducted using human saliva #1, which bound all the VLPs (Fig. 3b), as a source of HBGA carbohydrates. Briefly, saliva supernatant (1:200 in 1×PBS, pH 7.4) was coated on 96-well "U" bottom vinyl plates (Costar) overnight at $4\,^{\circ}C$ and blocked for 1 h at room temperature with 5% blocking buffer (Bio-Rad). Another 96-well "U" bottom vinyl plate was blocked overnight at $4\,^{\circ}C$ and used to incubate duplicates of two-fold serial dilutions of serum samples (1:100 in 5% blocking buffer) with GII.17 VLPs at 0.25–0.5 μg/ml (depending on affinity of VLPs to the HBGAs in the saliva) for 1 h at $37\,^{\circ}C$. Pre-incubated serum-VLPs mixture was then added to saliva-coated plates and incubated for additional 1 h at $37\,^{\circ}C$. 1:5,000 dilution of pooled serum from guinea pigs immunized with different GII.17 VLPs were used as primary detection antibodies against bound VLPs. Signals of binding (or blocking of binding) of VLPs on HBGA carbohydrate was determined using goat anti-guinea pig IgG-HRP (SeraCare, Cat#5220-0366, 1:2,000 dilution) and ABTS (SeraCare). The $OD_{405nm}$ values of serum dilutions were measured using SPECTROstar Nano plate reader (BMG LABTECH). The $EC_{50}$ values, measured by reciprocal of serum dilution, were calculated from the normalized $OD_{405nm}$ curves and visualized using GraphPad Prism v10.

### Antigenic distance analyses

The blockade $EC_{50}$ titer table generated in HBGA-blockade assay was transformed into two-dimensional antigenic map using antigenic cartography[37] as implemented in Racmacs package v1.2.9 in R v4.5.0. The titer data was well projected on the map with strong correlation between titer table distance and map distance (correlation coefficient = 0.859, Supplementary Fig. 7a) after reaching to convergence during 500 optimizations (Supplementary Fig. 7b). The sample uncertainty was assessed using bootstrap method and visualized as blobs on the map (Supplementary Fig. 7c). As expected from the blockade $EC_{50}$ titer table, the two unique viruses (Tokyo27-3/Japan/1976 and Arg13099/Argentina/2015) showed larger blob size and uncertainty of their positions on the map. Those with low intra-cluster titers (CA-RGDS-1180/US/2023 and Arg4446/Argentina/2005) also presented relatively larger uncertainty. This uncertainty was reviewed using three-dimensional projection of the titer table (Supplementary Fig. 7d) and we confirmed that there was no significant difference in locations and

pair-wise antigenic distance between two- and three-dimensional maps (Supplementary Fig. 7e). Therefore, while there was uncertainty of the projection, especially for the coordinates of unique viruses, we used antigenic distance measured on the two-dimensional map as the map in general represents well the antigenic relationship observed in the $EC_{50}$ titer table (Fig. 4a and b). The antigenic distance among GII.17 VLPs were extracted and used to associate with their genetic relationship using R v4.5.0. First, Pearson's correlation between antigenic distance and genetic distance (the number of amino acid differences) was calculated using phangorn v2.12.1 and stats v4.5.1 packages in R v4.5.0. To identify sites contributing to the strong correlation, the correlation-scanning analysis was performed, in which correlation coefficient between antigenic distance and genetic distance within 20 amino acids sliding windows (step size of 5 amino acids) were calculated and visualized. To further identify potential antigenic sites, we applied machine-learning algorithm and random forest regression analysis using randomForest package v4.7-1.2 in R v4.5.0. Briefly, genetic distance was calculated at individual positions on the VP1 amino acid sequences and associated with the antigenic distance of the pairs of the viruses. We compared two different distance matrices: (i) simply, the number of changes between the pair, which is 0 for same residues and 1 for different residues used, and (ii) genetic distance calculated using Blosum62 substitution model as implemented in phangorn package v2.12.1 in R v4.5.0, which provides more continuous values of differences based on the physicochemical similarity of amino acid types used. Both two matrices provided good fit to the antigenic distance as indicated by %variation explained >90 in full dataset (all pairs-included) and >87 in subset only including clusters C, D, and new viruses in 100 runs of random replications (Supplementary Fig. 15). As the Blosum62-based genetic distance provided slightly better $R^2$ values (p < 0.0001 in two-sided t-test), we used this model to further evaluate the data. Top 30 important features (positions) in the predictive models using data from all or C/D/new cluster viruses (%Variation explained >88 and mean squared residuals <0.8) were then extracted (Fig. 4e and sequence alignment in Supplementary Fig. 8) and visualized on the P domain structure using USCF ChimeraX v1.9 and PDB #5F4M (Kawasaki323/Japan/2014 from cluster C virus). Of note, correlation of antigenic and genetic distance among viruses from cluster A and B was relatively low, i.e., r < 0.75 in all positions (Fig. 4d, middle panel), and we could not build a good predictive model using viruses only from A and B (%Variation explained = -54.03 and mean squared residuals = 3.65). Additionally, the performance of this model was confirmed using genetic and HBGA-blockade assay data obtained from GII.4 norovirus in previous study[38] (Supplementary Fig. 9).

### Seroprevalence of GII.17 noroviruses using human serum or plasma samples

To determine the seroprevalence against different GII.17 noroviruses in the human population, we titrated anti-GII.17 VLPs antibodies in human serum or plasma samples using ELISA and HBGA blockade assays. Three sets of human serum and plasma samples (pre-epidemic phase, 2021–2022; pre-US epidemic phase, August 2024; and post-epidemic phase January 2025) were obtained from adult donors through the NIH Blood Bank or Innovative Research (catalog # ISERS10ML), without information regarding prior norovirus infection history. Samples from NIH Blood Bank were collected from healthy volunteer donors at age ≥18 years old (ClinicalTrials.gov ID: NCT00001846). All human samples were collected in the United States. The 96-well "U" bottom microtiter plates (Thermo Scientific) were coated with 0.5 μg/ml VLPs in 1×PBS (pH 7.4) (Gibco) overnight at 4 °C and blocked for 1 h at room temperature with 5% blocking buffer (Bio-Rad). Human serum or plasma samples (1:50, fivefold serial dilution) were plated in duplicate and incubated with VLPs for 2 h at room temperature. Plates were washed three times with 1×PBS, 0.1% Tween 20 (Sigma Aldrich). Binding antibodies on VLPs were incubated with

goat anti-human IgG-HRP (SeraCare, Cat#5220-0330, 1:2000 dilution) for 1 h at room temperature. After three washes, ABTS was added to measure $OD_{405nm}$ values. Measured values were summarized using AUC using GraphPad Prism v10. HBGA-blockade assays using human samples were similarly conducted as described above. Because most samples presented blockade titers less than the detection limit ($EC_{50}$ < 25), statistic test was performed only to the IgG ELISA data using One-way ANOVA and Tukey multiple comparison test using GraphPad Prism v10. Normality was confirmed using Shapiro-Wilk test. To compare the epidemiological trend of GII.17 infections in the United States, surveillance data was downloaded from CaliciNet Data (https://www.cdc.gov/norovirus/php/reporting/calicinet-data.html) on March 14, 2025.

### Ethics statements

The collection and de-identification of samples and sequence data was conducted as part of national surveillance programs in Germany, Spain, and Argentina as described below.

In Germany, data of molecular characterization of noroviruses were analyzed on the basis of routine national infectious disease surveillance duties by local and state health departments and the Robert Koch Institute as laid out in the German Infection Protection Act. Thus, a review by an ethics committee was not required. In accordance with §13 of the German Infection Protection Act, laboratories are permitted to send patient samples to national reference centers and consultant laboratories for further analysis.

In Spain, data and sample collection was based on the routine molecular characterization activities for gastroenteritis viral outbreaks coordinated by the Instituto de Salud Carlos III through the National Center for Microbiology and available to all autonomous communities in Spain. This activity is carried out within the framework of the Spanish General Public Health Law (Law 33/2011) and Royal Decree 568/2024, which provide the legal basis for national public health surveillance and regulate the collection and analysis of epidemiological and microbiological data for public health purposes. Therefore, specific ethical approval was not required.

In Argentina, the study was conducted using samples collected through the National Surveillance Program for Norovirus and Rotavirus in Argentina, coordinated by the Viral Gastroenteritis Laboratory of the Instituto Nacional de Enfermedades Infecciosas (INEI–ANLIS "Dr. Carlos G. Malbrán"). All samples were analyzed within the official framework of public health surveillance from the Ministry of Health and were anonymized prior to testing to ensure patient confidentiality and privacy. Therefore, approval by an ethics committee was not required in accordance with current national regulations (Resolution 1480/2011 Section B, and 1715/2007, and Personal Protection Data Law # 25.326).

### Reporting summary

Further information on research design is available in the Nature Portfolio Reporting Summary linked to this article.

## Data availability

The generated fasta files and fastq files obtained in this study were deposited in the GenBank and SRA (BioProject: PRJNA1279050), respectively. Accession numbers are listed in Supplementary Data 1 and 5. Accession numbers of publicly available sequences retrieved from GenBank that were analyzed in this study are listed in Supplementary Data 2. Source data are provided with this paper and available at FigShare (https://doi.org/10.6084/m9.figshare.29421056).

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

## Acknowledgements

We acknowledge the Sequencing Core Facility of the Genome Competence Center (MF 1) at the Robert Koch Institute and the Genomics and Bioinformatics Departments, as well as technicians (Nerea Garcia-Ibañez, Juan Camacho, and Estrella Ruiz de Pedro) at ISCIII, Spain, for their excellent sequencing services and technical assistance. We also thank the communities and surveillance teams in Germany, Spain, and

Argentina that provided samples as part of the norovirus outbreak surveillance system. We thank the animal technicians and veterinarians at the Division of Veterinary Services, Food and Drug Administration (FDA) for their support with the animal experiment. We thank Cara J Lepore for her technical assistance to produce VLPs. We thank Julia Enkelmann and Mirko Faber at Department of Epidemiology of Infectious Diseases, Robert Koch Institute and Juan I. Degiuseppe at Laboratory of Viral Gastroenteritis, INEI-ANLIS "Dr. Carlos G. Malbrán" for providing valuable discussions and reviews during the study. We thank all colleagues that submit their sequences into the public repositories so that the genetic analyses can be conducted in this study. Financial support for this work was provided by the FDA intramural funds (program number Z01 BK 04012 LHV, to G.I.P.) and by the FDA, Office of Counterterrorism and Emerging Threats (OCET), Medical Countermeasures Initiative (MCMi) (grant OCET 2024-0373, to G.I.P.). M.L. and K.A.P. are supported by a CBER, FDA-sponsored Oak Ridge Institute for Science and Education fellowship. Sequencing service at the Robert Koch Institute was partially funded by internal resources and the automation used in this study was supported by co-funding from the European Union's EU4Health program under Project No. 101113012 (IMS-HERA2, to S.N.). The sequencing of norovirus strains in Spain was partially supported by internal funding from ISCIII (grant PI23CIII/00009, to M.D.F-G.). The data and comments presented here are part of our research and do not bind or obligate FDA.

## Author contributions

K.T., S.N., and G.I.P. conceived the project; S.N., S.J., B.A., M.D.F-G., and K.A.G. collected samples, screened for norovirus, and sequenced viral genomes; K.T. and I.M. conducted genetic analyses; K.T., J.A.K., and S.C. conducted literature meta-analysis; J.A.K. and K.T. curated genome database; K.T. and S.C. designed mutant VLPs; K.T., L.A.F-S., J.A.K., and G.I.P. generated VLPs; Y.G. conducted electron microscopy analyses; K.T. and L.A.F-S. conducted animal experiments; M.L., J.A.K., W.E. D., K.T., L.A.F-S., and K.A.P. conducted immunoassays; K.T. conducted bioinformatics analyses; G.I.P., S.N., and M.D.F-G. acquired funding; G.I.P. supervised the research activity; K.T. and G.I.P. wrote the original draft; and all authors reviewed and edited the paper.

## Competing interests

The authors declare no competing interests.
