## [Transparent Peer Review file · Nature Communications]

GII.17 norovirus re-emerged in the 2020s as a result of dynamic and adaptive evolutionary processes

Corresponding Author: Dr Gabriel Parra

Version 0:

Reviewer comments:

Reviewer #1

(Remarks to the Author)

Noroviruses have long remained a leading global pathogen causing acute diarrhea, thanks to their adaptive evolution. Over the past two decades, GII.4 has consistently been the predominant genotype of noroviruses. In 2014-2015, GII.17 suddenly emerged in Asian regions and attracted widespread attention, primarily due to the appearance of the new GII.17 Kawasaki 2014 variant. However, its prevalence quickly dropped to a low level. In recent years, the prevalence of GII.17 has risen again, which is believed to be associated with the emergence of another new GII.17 variant.

This paper focuses on the re-emergence of GII.17 noroviruses and reveals the mechanisms underlying the formation of their transmission advantages through genomic analysis, experimental validation, and evolutionary dynamics studies. These mechanisms include enhanced host-binding ability, antigenic variation, and selection of phenotypes enabling efficient transmission, providing crucial basis for norovirus vaccine development and prevention/control strategies. The study features a mature research approach and substantial results, holding significant importance for an in-depth understanding of the evolutionary patterns of noroviruses.

The study also has certain limitations, which the authors have elaborated on. Additionally, the following comments are put forward:

1. It is recommended to refine the title of the article. Although the authors have conducted sequencing and analysis of the entire genome, the research on adaptability and dynamic evolution is centered around the capsid protein VP1.
2. In Lines 138-139 and Figure 1a, what is the statistical method used for the number of reported GII.17 infections?
3. The VP1 sequences analyzed in the article use a length of over 1200bp as the screening criterion. What is the basis for this?
4. Figure 3, the impact of polymerase mutations is evaluated by examining differences in viral load in samples. However, factors such as host status and sampling time may affect viral load. Have the authors considered these factors?
5. The results in Figure 5f only show that infections with GII.17 and GII.4-2012 genotypes after 2024 led to seroprevalence in the host, but fail to reflect the antigenic diversity of GII.17 noroviruses.
6. Regarding the dynamic evolutionary process of GII.17, the authors investigated the diversity in 49 individuals using high-throughput sequencing. How were these 49 individuals selected? Generally, individuals with chronic infections are more conducive to the generation of viral diversity.
7. In Lines 147-148 and Line 405, the authors mention that the GII.17-B variant has been circulating cryptically since at least 2018. Unlike GII.4, where previous variants basically no longer appear after being replaced by new ones, GII.17 exhibits the phenomenon of multiple variants coexisting. What are the authors' insights into this?
8. It is recommended to further improve and enrich the results and related discussions on antigen prediction using machine learning.
9. The authors examined the diversity and adaptive evolutionary processes of GII.17 noroviruses from the perspectives of global, local population, and individual host levels. What does the author consider to be the differences in evolutionary characteristics between the GII.17 genotype and the GII.4 genotype?
10. Recombination is crucial for the prevalence of noroviruses, and it is recommended to supplement relevant discussions.

Reviewer #2

(Remarks to the Author)

In the manuscript titled "GII.17 norovirus re-emerged as a result of dynamic and adaptive evolutionary processes", Tohma et

al. investigated the diversification patterns, mutations patterns and intra-host viral diversity and adaptation processes using a panel of bioinformatic tools based on a large amount of data. The manuscript is written in a clear and concise fashion. The data were thoroughly analyzed to support their conclusions and the limitations of the study were listed. The genetic and antigenicity analysis based purely on VP1 surely can not answer the sudden emergence of the novel GII.17, but the results provided a glimpse of advantage gained in the evolution such as broadened binding to non-secretors. I only have a few questions when addressed might improve the accuracy of the manuscript.

1. In figure 2f, there is a documented GII.3 strain isolated in 2011 with a length of 549 aa for VP1 (GenBank accession number, AGI17594).

2. Please provide the amount of VLPs used for mouse immunization.

3. In figure 5a, if my understanding is correct, the cells in white indicate that the serum showed no blocking effects at the starting dilution of 1:100. This is in stark contrast with GII.2, GII.3 and GII.6 norovirus based on our experience. We have experienced variations in blockade assay using different saliva samples, so have the authors tried other saliva samples other than sample 1?

4. The authors used EC50 to reflect the antigenic distance of different GII.17 variants rather than the cross-reactive IgG titers. Based on our experience, blocking antibodies only accounted for a small part of the antibodies produced during immunization and this can be easily observed during monoclonal antibody production that only limited number of cell clones show blocking effects.

Reviewer #3

(Remarks to the Author)

The manuscript "GII.17 norovirus re-emerged as a result of dynamic and adaptive evolutionary processes" by Tohma et al. focus on GII.17 noroviruses. This study merges two distinct research perspectives on GII.17 noroviruses. First, it analyzes active surveillance data from Germany, Spain, and Argentina collected during the GII.17 surge between 2021 and 2024 to highlight the re-emergence of these viruses and investigate in-depth mutational patterns that may have driven this surge in these regions. Second, it conducts a comprehensive analysis of publicly available GII.17 sequences from all time periods to understand global diversification patterns and study the dynamic and adaptive evolution of these viruses. While extensive, the combination of these broad aims dilutes the study's overall objective, making its central purpose unclear.

In my opinion, the strains from the active surveillance data from Germany, Spain, and Argentina provides new information aligned with the current surge of GII.17 viruses and those should be the central focus of this study and data should be rearranged accordingly for in-depth mutational analysis, strain diversification patterns and antigenic characteristics and differential HBGA binding profile of these viruses with primary focus on GII.17 strains from studied geographic region(s).

The second approach, a comprehensive analysis of publicly available GII.17 sequences, exhibited significant redundancy. Much of the older strain data reported before 2021, particularly concerning clusters A, B, C, and D (including HBGA binding), has been adequately covered in previous studies. Replicating similar experiments on known clusters and re-analyzing previously published information provides minimal new insight into pre-2021 GII.17 strains. The authors should prioritize the analysis of the current GII.17 strain surge and reorganize their data and discussion to reflect this focus.

The manuscript contains excessive detail and redundant information (both textual and illustrative), which hinders readability. Condensing the text to focus solely on the study's key findings and significant contributions is recommended. Details already established in prior research should be removed, while other less crucial information could be moved to supplementary materials.

Title needs improvement for clarity: GII.17 noroviruses re-emerged where (Germany, Spain and Argentina? OR worldwide?) and when? (2021-2024?).

The authors consistently refer to GII.17 phylogenetic clusters as "variants" throughout the manuscript. However, a formal classification system for GII.17 variants has yet to be established. This paper's objective is not to propose an updated classification. It is understood that the ad hoc international norovirus nomenclature working group (Kroneman et al., 2013 and Chhabra et al., 2019), is responsible for assigning names and numbers to norovirus genotypes and variants. It is crucial to determine whether the presented data has been analyzed, discussed, or approved by this group. Also, as previous norovirus classification updates suggest, the naming of variant types should ideally be based on the geographic location of the first identified strains, such as GII.4 New Orleans or GII.4 Sydney. The current use of names like A, B, C, and D warrants clarification as to whether these have been officially approved by the working group. Authors are advised to consider using the term "clusters of interest" in the absence of a properly approved classification system. This approach maintains scientific rigor while awaiting formal guidelines from the designated nomenclature body.

In the introduction section, references [15–21] describe GII.17 variants as the Kawasaki-308 cluster, Kawasaki-like, or Kawasaki-2014 like cluster, not using the A-D and "new" designations. To ensure clarity for readers, authors should either utilize publications that employ the A-D nomenclature for GII.17 clusters (e.g., PMID: 26196235; 40476850), or explicitly link the A-D clusters to their commonly recognized names. The 'new' cluster (2021-2014) is most commonly referred to as the Romania-2021 like cluster in recent literature (e.g., PMID: 39328162, 40576844). Authors are advised to align with recent reports to prevent confusion and facilitate the tracking of these viruses.

Figure 1 can be omitted, as most of its content is already available in previously published work and can be cited accordingly (e.g., PMID: 40476850). Duplicating this information should be avoided.

Lines 135-158 present previously established information rather than novel results. Authors are advised to remove this

material from the results section.

Lines 163-175: To better understand..... decades (Figure 2c). The VP1 amino acid-based phylogenetic tree clearly illustrates the genetic distance between the clusters. However, it is unclear what additional information, beyond cluster distances and the number of amino acid changes, is conveyed by the two-dimensional (2D) sequence space map. For conciseness, authors are requested to streamline this section, moving detailed explanations to the supplementary materials.

Lines 176-177: 'a similar pattern shown by GII.4 pandemic viruses 33, 34.' This can be discussed but it cannot be a part of results of the study.

Lines 185-192: This is previously known information Authors are advised to remove this material from the results section. This can be a part of discussion.

Lines 207-215: 'Interestingly, the GII.P17..... (Supplementary Figure 4). This new cluster, positioned between GII.P17 and GII.P3, has been previously studied (PMID: 39328162). That prior research concluded that this cluster (thus far identified solely in the United States) remains within GII.P17. Supplementary Figure 4, which depicts the sub-lineage through a phylogenetic tree, provides no new information. Authors are advised to either condense this information or remove it from the results section.

Lines 207-2012: Lack of supporting data. Please reword or remove.

Recombination analysis: A) Recombination in noroviruses is frequently observed within the ORF1/ORF2 region. GII.4 serves as a prime example, where intra-genotype (variant or cluster level) recombination between ORF1 and ORF2 is common. The established dual genotyping or variant typing system largely negates the need for SimPlot analysis to examine ORF1/ORF2 combinations, especially at the intra-genotype level. Additionally, a variant-level classification for GII.17 ORF1 remains undefined. Therefore, please consider removing the figure and condensing the accompanying text. B) The Oregon112 strain originates from the United States, outside the geographical scope of this study. This strain's ORF1 and ORF2 have already been analyzed in a previous study conducted in the United States (PMID: 39328162). To maintain the study's focus, the genome-wide mutation analysis should prioritize the strains identified in the present investigation.

HBGA binding to human sera and the study of host cell attachment factor is a complete study in itself with another research prospective. Authors efforts to combine the study with rest of the analysis resulted in overinterpretation of the correlation between predicted antigenic residues and observed epidemiological patterns as there are significant evidence gaps that have not been addressed experimentally. Critical methodological components, site directed mutagenesis, and reverse genetics confirmation of machine-learning predictions, are notably absent. Although the work is described as a "comprehensive antigenic characterization," its scope is not matched by experimental breadth or validation. This section need rewriting. The complete section is overly narrative in tone, diffuse in focus, and lacking the experimental substantiation needed to firmly support its conclusions.

Minimizing extraneous details, this section should concentrate on delineate the significant new information on recent emerging GII.17 strains. Additionally, prior investigations have thoroughly examined HBGA interactions for GII.17 strains identified before the recent surge. To maintain relevance and impact, the analysis presented here, and the subsequent discussion, should primarily address the current GII.17 surge.

Version 1:

Reviewer comments:

Reviewer #1

(Remarks to the Author)

The author has provided a detailed and thorough response to my comments, and I have no further supplementary remarks at this time. The article conducts a comprehensive and in-depth study on the evolutionary mechanism of the epidemic caused by the new norovirus variant, and I agree to its publication.

Reviewer #2

(Remarks to the Author)

Due to lack of data sharing among labs, the conclusions in the manuscript might have been understated. In fact, the current reported novel GII.17 strains have been circulating in mainland China during the 2024/2025 winter season. VP1 amino acid sequence of 100% similarity to the reported novel GII.17 has been isolated and sequenced, and the complete genome sequences will be available in the NCBI database soon. The possible already widespread transmission in the inland of China and the ratio of non-secretors in the population (approximately 20%) warrant further investigation of the role of extra binding ability to HBGAs. The authors have mentioned this in their limitations. I wish more excellent work from your lab.

Reviewer #3

(Remarks to the Author)

The revised manuscript "GII.17 norovirus re-emerged in the 2020s as a result of dynamic and adaptive evolutionary processes in its major capsid protein" by Tohma et al. focusses on the current surge of GII.17 noroviruses in Germany, Spain, and Argentina and understanding the genotypic and phenotypic characteristics of noroviruses obtained in this study. While the major strengths of this study include the genomic data from the recent GII.17 viral outbreaks in Germany, Spain, and Argentina; and the machine learning/ binding study using a large panel of VLPs to account for genomic differences both within and between known GII.17 clusters. I also appreciate the authors' effort to accommodate all three reviewers' comments, which has made this version relatively clear. After carefully reviewing the rebuttal and the updated manuscript, I have the following concerns:

1. The study effectively outlines its primary objectives in the reporting summary. However, the authors' attempt to provide a comprehensive overview of the GII.17 story, by including previously published information, weakens the overall impact of this study, particularly in the sections discussing genomic variations.

For instance, the two Results sections, "Substitutions, insertions, and deletions play a role in shaping the diversity of GII.17 norovirus" and "Unique ORF1 and ORF3 genes contribute little to viral replication," read more like a literature review and in their current form do not present any new findings. The authors acknowledge that the information presented has been previously reported or explained in other studies, which is not appropriate for the Results section of an original research manuscript.

The Results section should focus on the novel findings derived from the current study's data, with citations and comparisons reserved for the Discussion section. Specifically, these two sections should prioritize the strains examined in this study (from Germany, Spain, and Argentina) as they represent the new information regarding the recent GII.17 surge. All other information in this section should be presented in relation to these specific strains (identified in this study).

2. Furthermore, to provide a comprehensive overview of the GII.17 noroviruses, important details about the data generated from this study are missing that make the genomic variation story difficult to follow:

a. Please clarify the composition of (a) the 1,400 GII.17 sequences. Was the breakdown 889 archival (are these all GenBank downloads, and if so, when was the last download performed, or were other sources used?), and (b) the composition of the 511 new sequences from surveillance in Germany, Spain, and Argentina?

b. Please specify the sequence type, whether full genome, partial, VP1 only, P2 only, for each of the 1,400 sequences. This detail needs to be added to Supplementary Table 1. A new supplementary table should also list all archival sequences used.

c. Please state how many strains (both archival and new) were used for the analysis from each specific region (ORF1, ORF2, P2, and ORF3).

d. Currently, there is a discrepancy in the numbers presented. The Results section (lines 144-146) mentions 1,400 genomes for Figure 1b, but the legend for Figure 1b shows 1,013 sequences. Authors should correct this.

e. There were 49 full genome sequences generated in this study. Figure 1b highlights 68 VP1 sequences. Please specify the number of VP1 sequences generated in this study.

f. Another inconsistency exists with the archival sequence count. If 511 sequences are new, the archival count should be 889 (1400 - 511). However, the legend for Figure 1b states 945 archival sequences. This needs to be explained/corrected.

3. Figure 1b: Please clarify why a 1500-nt sequence was chosen for the VP1 analysis instead of the complete sequence. Only five strains were incomplete, the reason for not removing them from the analysis or excluding them from the complete VP1 dataset should be explained. If these five strains are critical for the analysis, please justify their importance. Specify which end of the VP1 sequence was incomplete (3' or 5'). Explain in the Discussion section, how the missing sequence data might affect the phylogenetic analyses and how it would potentially influence the identification of new clusters associated with the current surge.

4. Figure 1b is the main figure of this study. Authors should specify whether amino acid or nucleotide sequences were used to construct the VP1 phylogenetic tree in Figure 1b. If nucleotide sequences were used, the authors are advised to regenerate the tree using amino acid sequences to be consistent with the literature for comparison and alignment with recommended criteria for new cluster identification.

5. Lines 173-175: Consider moving this line in the discussion.

6. Lines 193-194 and figure 2a: I don't see any phylogenetic divergence in figure 2a that warrants placement of new GII.17 cluster(s) in a new or any known polymerase type. The phylogenetic tree in figure 2a is similar to as shown in reference 28. Authors are advised to please remove or change the text in line 193-194.

7. Lines 198-204: This is not new information. The sub-clustering of polymerase sequences from recent GII.17 strains has already been described in reference 28 as sub-lineage I and II. To avoid literature confusion, please be consistent with already known information while referring to these sub-lineages.

8. Abstract lines 54-56: This statement is true only for the study regions. European countries experienced the GII.17 surge earlier (2023-24) compared to the Americas (2024-2025) and other geographic regions. Additionally, limited genetic diversity was observed among the GII.17 strains circulating in many of these regions. It is possible that sequences from geographic regions undergoing the 2025-26 GII.17 surge might present a different epidemiological picture. Please re-word these lines to avoid generalizing the findings to a global context.

9. A key strength of this study is the HBGA blocking assay. The revised sections have been improved and are now much clearer.

10. Lines 182-183: role of residues 393-396 in cluster D has been explained in earlier study (PMID: 28973354). Please discuss accordingly.

11. The confirmation of cluster-specific binding by the authors replicates previously published findings for GII.17 viruses (e.g., PMID: 28973354). While this study expands the scope by testing 16 VLPs against 9 different HBGA saliva samples, its main conclusion—that binding is cluster-specific—is identical to earlier reports using only 1-4 VLPs. This significant overlap with existing literature must be discussed more robustly.

Thank you very much for considering our manuscript (NCOMMS-25-49440) for publication in Nature Communications. We would also like to thank the reviewers as the manuscript has improved based on their comments.

We have revised the manuscript based on the comments from reviewers and the editor. In this revised version, we adopted the term “cluster” instead of “variant,” as recommended by reviewer #3. Additionally, we have included detailed description of the statistical tests used, in line with the editor’s request and to fulfill the requirements listed on the Reporting Summary. Please find the specific responses to the reviewers’ comments below.

Reviewer #1

Noroviruses have long remained a leading global pathogen causing acute diarrhea, thanks to their adaptive evolution. Over the past two decades, GII.4 has consistently been the predominant genotype of noroviruses. In 2014-2015, GII.17 suddenly emerged in Asian regions and attracted widespread attention, primarily due to the appearance of the new GII.17 Kawasaki 2014 variant. However, its prevalence quickly dropped to a low level. In recent years, the prevalence of GII.17 has risen again, which is believed to be associated with the emergence of another new GII.17 variant.

This paper focuses on the re-emergence of GII.17 noroviruses and reveals the mechanisms underlying the formation of their transmission advantages through genomic analysis, experimental validation, and evolutionary dynamics studies. These mechanisms include enhanced host-binding ability, antigenic variation, and selection of phenotypes enabling efficient transmission, providing crucial basis for norovirus vaccine development and prevention/control strategies. The study features a mature research approach and substantial results, holding significant importance for an in-depth understanding of the evolutionary patterns of noroviruses.

The study also has certain limitations, which the authors have elaborated on.

Response: We appreciate for your comments and suggestions. Please find below our responses to your specific comments.

Comments:

1. It is recommended to refine the title of the article. Although the authors have conducted sequencing and analysis of the entire genome, the research on adaptability and dynamic evolution is centered around the capsid protein VP1.

Response: Thank you for your suggestion. We changed the title to address your suggestion and that from Reviewer #3. The revised title now reads “GII.17 norovirus re-emerged in the 2020s as a result of dynamic and adaptive evolutionary processes in its major capsid protein”.

2. In Lines 138-139 and Figure 1a, what is the statistical method used for the number of reported GII.17 infections?

Response: We counted the number of publications (PubMed IDs) that reported at least one GII.17 sequence (GenBank accession number) detected in humans for each year. However, as suggested by

Reviewer #3 –and given that we recently summarized the epidemiological trends of historical noroviruses, including GII.17, in a review paper (Parra et al. J Gen Virol 2025 DOI: 10.1099/jgv.0.002118), we decided to omit Figures 1a and 1c from the manuscript.

3. The VP1 sequences analyzed in the article use a length of over 1200bp as the screening criterion. What is the basis for this?

Response: First, we apologize for the typographical error in our original manuscript. We collected sequences with a length of $\geq 1,500$ nucleotides, not $\geq 1,200$ nucleotides as previously stated. This criterion was designed to include as many sequences as possible in order to capture a comprehensive view of the evolutionary history of this genotype. The majority of VP1 sequences included in this study corresponded to a complete (1,617 – 1626 nt in length) VP1, with only five sequences with nearly complete VP1 sequences ($\geq 1,567$ nt in length). These five viruses belong to cluster D and were included in our analyses to minimize any potential dataset biases.

4. Figure 3, the impact of polymerase mutations is evaluated by examining differences in viral load in samples. However, factors such as host status and sampling time may affect viral load. Have the authors considered these factors?

Response: We agree with this reviewer that sampling biases could be introduced by differences in sample collection, timing post-infection, and host-related factors (e.g., immunity, HBGA profiles). Despite this possibility, we chose to include this data for the following reasons: (i) the samples were collected as part of routine surveillance of outpatients presenting symptoms -typically within 48 hrs of infection, when viral titers peak; (ii) the same RNA extraction and RT-qPCR protocols were used to quantify the viral genome copies across all samples included; and (iii) our analysis included over 600 samples. Therefore, we expect that the large sample size and consistent methodology help minimize the effects of these potential sources of variability among samples. However, because only limited metadata (sampling location and date) were available for each sample and we could not control for the factors mentioned above, we have now explicitly acknowledged this limitation in the Discussion section of the revised manuscript (Page 23 Line 508-510).

5. The results in Figure 5f only show that infections with GII.17 and GII.4-2012 genotypes after 2024 led to seroprevalence in the host, but fail to reflect the antigenic diversity of GII.17 noroviruses.

Response: The human serum samples were collected from healthy donors. We do not know their infection histories, but they were not convalescent serum. We detected blockade antibodies from only a few individuals and therefore, we also included ELISA IgG titers (upper panel), which are more sensitive but can cross-react with different variants and even genotypes (see Ford-Siltz et al. J Infect Dis 2022 DOI: [10.1093/infdis/jiaa116](https://doi.org/10.1093/infdis/jiaa116)). While HBGA blocking antibodies generally map to variable antigenic sites (Tohma et al. Cell Rep 2022 DOI: [10.1016/j.celrep.2022.110689](https://doi.org/10.1016/j.celrep.2022.110689)), ELISA detects all the antibodies including those mapping to conserved regions of the capsid. Although serum samples with high level of antibodies can still differentiate homotypic and heterotypic responses using ELISA (see our response to reviewer #2, comment 4), the absence of convalescent-phase samples, when anti-norovirus antibody responses peak, likely explains the limited ability to distinguish between GII.17 viruses. Notably, while all the variants presented statistically significant increase of IgG titers over time, those against Gaithersburg (cluster D) and 144 (new cluster) presented largest increases of titers as compared to Kawasaki323 (cluster C) and other minor viruses. This pattern may reflect an anamnestic response due to previous infections with cluster D viruses. This was discussed in Page 15 lines 330-332 and Page 22 lines 489-490.

6. Regarding the dynamic evolutionary process of GII.17, the authors investigated the diversity in 49 individuals using high-throughput sequencing. How were these 49 individuals selected? Generally, individuals with chronic infections are more conducive to the generation of viral diversity.

Response: The selection of the 49 positive samples was done randomly from those with sufficient material available for full-genome NGS analysis, with the aim of representing different years and geographic locations (federal states in Germany). Thus, the selection included: 4 samples collected in 2021, 13 in 2022, 11 in 2023, and 21 in 2024 (Supplementary Table 5). We added this information in the Methods section (Page 24 Line 538). These samples were collected from norovirus positive cases reported during gastroenteritis outbreaks or sporadic infection by local public health authorities in Germany. Patient information is anonymized, and we do not have information on host immune status. Regarding chronic infections, these typically occur in individuals with varying degrees of immune deficiency or suppression. Indeed, such individuals often exhibit greater viral diversity; however, this diversity is usually associated with the presence of two or more distinct haplotypes (i.e., clonal viral populations) (Izquierdo-Lara et al. *EBioMedicine* 2024 DOI: 10.1016/j.ebiom.2024.105391; Chaimongkol et al. *mBio* 2023 DOI: 10.1128/mbio.02177-23; Venturini et al. *Virus Evol* 2022 DOI: 10.1093/ve/veac093). This was not observed in any of the 49 individuals included in this study.

7. In Lines 147-148 and Line 405, the authors mention that the GII.17-B variant has been circulating cryptically since at least 2018. Unlike GII.4, where previous variants basically no longer appear after being replaced by new ones, GII.17 exhibits the phenomenon of multiple variants coexisting. What are the authors' insights into this?

Response: This is a very interesting question and a subject of ongoing research of our group. However, we do not yet have a definite answer at this time. There are multiple genotypes that present genetically distinct variants (or clusters), such as GI.3, GII.6, GII.17, that have been cocirculating in the population (Parra et al. *PLoS Pathog* 2017 DOI: 10.1371/journal.ppat.1006136; Tohma et al. *J Gen Virol* 2018 DOI: 10.1099/jgv.0.001088), which contrast to the variant (or cluster) replacement of GII.4. Notably, the evolutionary tree of GII.4 exhibits a ladder-like structure, resembling that of influenza virus, suggesting continuous lineage replacement over time. In contrast, the trees of non-GII.4 genotypes are characterized by long branches, indicative of extended divergence times and limited ongoing transmission. Cross-reactivity of evolutionary adjacent GII.4 variants could partly explain their epidemiological patterns associated with host immune pressure and herd immunity (Lindsmith et al. *PLoS Med* 2008 DOI: 10.1371/journal.pmed.0050031; Kendra et al. *PNAS* 2021 DOI: 10.1073/pnas.2015874118); however, this information is not currently available for non-GII.4 noroviruses. Indeed, this study represents the first large-scale exploration of antigenic diversity for a norovirus genotype other than GII.4. We speculate that the coexistence of multiple variants could be attributed to several factors: (i) low level of host susceptibility, such as narrow HBGA binding profiles in GII.17 clusters A and B, resulting in low prevalence; (ii) distinct antigenic profiles that prevent herd immunity from other variants to induce antigenic pressure; (iii) strong fitness constraints that limits antigenic diversity and host susceptibility. Notably, when major evolutionary events occur, e.g. the introduction of insertions and deletions, some of these can provide new fitness landscapes resulting in the emergence of new variants (or clusters). These events have been described in this study and in our GII.4 study (Tohma et al. *Cell Rep* 2022 DOI: 10.1016/j.celrep.2022.110689). A working model that we hope will prompt other colleagues to examine all these factors have been presented in our latest review (Parra et al. *J Gen Virol* 2025 DOI: 10.1099/jgv.0.002118).

8. It is recommended to further improve and enrich the results and related discussions on antigen prediction using machine learning.

Response: We appreciate your comments and suggestions. We have now expanded these analyses and presented new analyses using GII.4 data from our previous paper (Kendra et al. PNAS 2021 DOI: 10.1073/pnas.2015874118). Using the same algorithms and similar data, i.e. large HBGA blocking titers datasets, we demonstrated that the machine learning models accurately predict the role of residues that have been experimentally demonstrated to play a role in the antigenic diversity of GII.4 noroviruses (e.g., Tohma et al. mBio 2019 DOI: 10.1128/mBio.02202-19; Lindesmith et al. J Virol 2012 DOI: 10.1128/JVI.06200-11). Although we commented in the Discussion section that a limitation of this study is that we did not validate the prediction using site-directed mutagenesis and mAbs, the new data related to GII.4 support our current analyses. At this time, we prefer not to include any speculative discussions regarding potential antigenic sites or epitopes of GII.17 norovirus, as experimental validation is still ongoing and forms an additional line of investigation in our laboratory for GII.17 and other genotypes. We hope that the additional discussion and information provided adequately address the reviewer's request. The new data related to GII.4 is now presented in Supplementary Figure 9.

9. The authors examined the diversity and adaptive evolutionary processes of GII.17 noroviruses from the perspectives of global, local population, and individual host levels. What does the author consider to be the differences in evolutionary characteristics between the GII.17 genotype and the GII.4 genotype?

Response: Again, this is a very interesting fundamental question about the biology of the different norovirus genotypes. At this point, our group is not sure if we actually understand all the biological characteristics that makes GII.4 the most prevalent virus. The current working model is that there is a chronological replacement of variants every 2-3 years that facilitates immune escape, but this pattern was only observed during 2002-2012 (Parra Virus Evolution 2019 DOI: 10.1093/ve/vez048). The last variant to emerge after this pattern was Sydney 2012 variant, which has been dominating for over 10 years without major genetic and antigenic changes (Parra et al. J Virol 2020 DOI: 10.1128/jvi.01716-22). Notably, during this time, other antigenically distinct variants have emerged, e.g., Hong Kong 2019, San Francisco 2018 (Chan et al. Emerg Infect Dis 2021 DOI: 10.3201/eid2701.203351; Chhabra et al. Emerg Infect Dis 2024 DOI: 10.3201/eid3001.231003; Tohma et al. Emerg Infect Dis DOI: 10.3201/eid3005.231694), yet they did not replace Sydney 2012. The pattern of rapid replacement of variants occurred after the insertion of amino acid 393 in the VP1 (Figure 1d), and a major change in the immunodominance of antigenic sites (Tohma et al. Cell Rep 2022 DOI: 10.1016/j.celrep.2022.110689), which could facilitate immune focus and viral escape (Greaney et al. Cell Rep Med 2021 DOI: 10.1016/j.xcrm.2021.100257). While we agree these are important topics, they fall outside the scope of the current study. These questions are an active focus of ongoing research in our laboratory, and we hope to provide more definitive insights in future work.

10. Recombination is crucial for the prevalence of noroviruses, and it is recommended to supplement relevant discussions.

Response: Thank you for your suggestion. Based on our (and others) analysis and data, it seems recombination was not a major factor in the emergence of the new GII.17 viruses. Rather, we found a lot of recurrent and parallel mutations on ORF1, which resulted in the ORF1 of this virus distant from previously circulated GII.P17 (variant C and D) viruses. Interestingly, some mutations in the "GII.P17" ORF1 region of recent GII.17 cluster B viruses place them intermediate on the tree between original GII.P17 and GII.P3, which was associated with previously circulated cluster B viruses. Understanding of the origins and mechanisms of these mutations warrant further study on

evolutionary patterns of ORF1. We expanded discussions on recombination accordingly (Page 19 Lines 422-427).

Reviewer #2

In the manuscript titled "GII.17 norovirus re-emerged as a result of dynamic and adaptive evolutionary processes", Tohma et al. investigated the diversification patterns, mutations patterns and intra-host viral diversity and adaptation processes using a panel of bioinformatic tools based on a large amount of data. The manuscript is written in a clear and concise fashion. The data were thoroughly analyzed to support their conclusions and the limitations of the study were listed. The genetic and antigenicity analysis based purely on VP1 surely can not answer the sudden emergence of the novel GII.17, but the results provided a glimpse of advantage gained in the evolution such as broadened binding to non-secretors. I only have a few questions when addressed might improve the accuracy of the manuscript.

Response: Thank you for your comments and suggestions. As requested by reviewers #1 and #3, we have now emphasized the results of VP1 on the title. Please find below our responses to your specific comments below.

Comments:

1. In figure 2f, there is a documented GII.3 strain isolated in 2011 with a length of 549 aa for VP1 (GenBank accession number, AGI17594).

Response: We sincerely appreciate your careful and thoughtful review of our data. We retrieved the dataset from our previous study, which excluded sequences from animals, environment, and immunocompromised individuals (Tohma et al. PLoS Pathog 2021 DOI: 10.1371/journal.ppat.1009744). The GII.3 virus (AGI17594, Hu/GII.3/NIHIC8.1/2011/USA) was collected from an immunocompromised patient as part of chronic norovirus infection surveillance conducted at NIH and therefore omitted from the analysis. We agree with this reviewer that this is indeed an interesting observation, particularly given that viruses with unique indels were reported exclusively during the chronic phase in immunocompromised patients (e.g., Chaimongkol et al. mBio 2023 DOI: 10.1128/mbio.02177-23; Lindesmith et al. J Virol 2019 DOI: 10.1128/JVI.01813-18). We clarified the selection criteria of the previous dataset in Figure legend (revised Figure 1d) and added comments on indels reported from immunocompromised patients in Discussion section (Page 20 Lines 440-444).

2. Please provide the amount of VLPs used for mouse immunization.

Response: We provided the amount of VLPs (40 µg VLPs/dose) used for mouse immunization in Methods section (Page 32 Line 708).

3. In figure 5a, if my understanding is correct, the cells in white indicate that the serum showed no blocking effects at the starting dilution of 1:100. This is in stark contrast with GII.2, GII.3 and GII.6 norovirus based on our experience. We have experienced variations in blockade assay using different saliva samples, so have the authors tried other saliva samples other than sample 1?

Response: Yes, that is correct -the white cells on the heatmap indicate the absence of blockade signals. The degree of cross-reactivity among different viruses depends on multiple factors, including immunization protocols, antigen quantity and adjuvant used, to obtain the hyperimmune serum, and the degree of genetic diversity among the viruses tested. In previous studies, we noticed stronger cross-blocking titers when using sera from guinea pigs immunized with VLPs and

CFA/IFA as adjuvant as compared with mice immunized with VLPs and Alum (Tohma et al. Cell Rep 2022 DOI: 10.1016/j.celrep.2022.110689; Ford-Siltz et al. J Infect Dis 2022 DOI: 10.1093/infdis/jiaa116; Kendra et al. PNAS 2021 DOI: 10.1073/pnas.2015874118). Moreover, degree of intra-genotype cross-reactivity could be varied depending on what genotype is under study. Both GII.2 and GII.3 do not present distinct genetic clusters (Tohma et al. PLoS Pathog 2021 DOI: 10.1371/journal.ppat.1009744) and therefore strong cross-reactivity could be expected among different strains within the genotypes. Indeed, Swanstrom et al. have shown that the cross-blockade titers remained the same using time-ordered GII.2 variants VLPs (Swanstrom et al. J Virol 2014 DOI: 10.1128/JVI.02793-13). GII.6 is divided into three major clusters (or variants) but only <10 mutations were accumulated within the clusters over 40 years (Parra et al. PLoS Pathog 2017 DOI: 10.1371/journal.ppat.1006136). Thus, cross-reactivity among different GII.6 noroviruses depends on which strains are being tested.

Regarding the last question on the variations derived from different HBGA carbohydrate sources, we have not conducted one-by-one comparison among different saliva samples. Instead, we tested a few VLPs and sera using saliva sample #1 and pig gastric mucin type III (PGM III) as a source of carbohydrates in the blockade assay (see Figure below, Panel a). PGM III contains H-, Le^y-, and A-antigens, but lacks Le^a and Le^b (Lindesmith et al. J Virol 2012 DOI: 10.1128/jvi.06200-11), which are the predominant HBGA carbohydrates present in our saliva sample #1. The overall titers were higher when using PGM III as compared with saliva #1, but there was no major difference in the trend of homotypic and heterotypic titers. As the titers were normalized during the antigenic cartography analyses, the difference of magnitude of titers did not result in significant changes on their positions on the antigenic map (Figure Panel b) nor antigenic distance (Figure Panel c). The overall distance was higher when using saliva #1, but the antigenic distance measured using saliva #1 and PGM III was strongly correlated with Pearson's $r = 0.9836$. Of note, we did not use PGM III in our large-scale blockade assays because VLPs from clusters A and B did not bind well to the PGM III to conduct blockade assays with them. In a previous study, Lindesmith et al. reported no significant difference of anti-GII.4 blockade titers when using different HBGA carbohydrate molecules (Lindesmith et al. PLoS Med 2008 DOI: 10.1371/journal.pmed.0050031). Thus, we do not expect major changes in the overall conclusions of our study when using different carbohydrate sources for the HBGA blocking assays.

Figure: Blockade titers of mouse sera raised against Kawasaki323 and Gaithersburg viruses using 5 VLPs from clusters C, D, and new viruses. (a) The heatmap summarizes blockade EC₅₀ titers using (left) saliva sample 1 and (right) PGM III as a source of HBGA carbohydrates. (b) Antigenic maps created using blockade titers from panel a (left: saliva sample 1, right: PGM III). (c) Correlation analysis of antigenic distance of VLPs (antigens). The x and y axes show the distance calculated from blockade titers measured using saliva 1 and PGM III, respectively. The Pearson's correlation coefficient r was 0.9836.

4. The authors used EC₅₀ to reflect the antigenic distance of different GII.17 variants rather than the cross-reactive IgG titers. Based on our experience, blocking antibodies only accounted for a small part of the antibodies produced during immunization and this can be easily observed during monoclonal antibody production that only limited number of cell clones show blocking effects.

Response: We agree that many antibodies are elicited against sites that are not involved in blocking of HBGA:VLPs interactions, and there could be neutralizing antibodies that do not map to HBGA binding sites (Alvarado et al. Nat Commun 2021 DOI: 10.1038/s41467-021-26418-1). However, the fact that (i) blockade titers are well correlated with viral neutralization titers at the mAb level (Tohma et al. Cell Rep 2022 DOI: 10.1016/j.celrep.2022.110689) and polyclonal level (Atmar et al. J Infect Dis 2024 DOI: 10.1093/infdis/jiae311; Atmar et al. J Infect Dis 2020 10.1093/infdis/jiz526), (ii)

strong blockade titers are associated with protection from diseases (Reeck et al. J Infect Dis 2010 DOI: 10.1086/656364; Atmar et al. Clin. Vaccine Immunol 2015 DOI: 10.1128/CVI.00196-15), and (iii) multiple non-blocking antibodies mapped on conserved S and P domains cross-react with genetically diverse viruses (Ford-Siltz et al. Microbiol Spectr 2024 DOI: 10.1128/spectrum.01143-24), indicate that antibody titers measured by ELISA do not reflect information of functionally relevant, protective antibodies. Regardless, we conducted ELISA experiments to demonstrate that it still differentiates viruses from the different clusters (see Figure below).

Figure: Blockade EC₅₀ titers (top panel) and ELISA AUC titers (bottom panel) of mouse sera raised against different GII.17 viruses from clusters A, B, C, and D.

Reviewer #3

The manuscript “GII.17 norovirus re-emerged as a result of dynamic and adaptive evolutionary processes” by Tohma et al. focus on GII.17 noroviruses. This study merges two distinct research perspectives on GII.17 noroviruses. First, it analyzes active surveillance data from Germany, Spain, and Argentina collected during the GII.17 surge between 2021 and 2024 to highlight the re-emergence of these viruses and investigate in-depth mutational patterns that may have driven this surge in these regions. Second, it conducts a comprehensive analysis of publicly available GII.17 sequences from all time periods to understand global diversification patterns and study the dynamic and adaptive evolution of these viruses. While extensive, the combination of these broad aims dilutes the study's overall objective, making its central purpose unclear.

Comments:

1. In my opinion, the strains from the active surveillance data from Germany, Spain, and Argentina provides new information aligned with the current surge of GII.17 viruses and those should be the central focus of this study and data should be rearranged accordingly for in-depth mutational analysis, strain diversification patterns and antigenic characteristics and differential HBGA binding profile of these viruses with primary focus on GII.17 strains from studied geographic region(s).

Response: We appreciate reviewer’s valuable comment but respectfully disagree. The primary objective of this study was to investigate the mechanisms underlying the recent emergence of new GII.17 noroviruses, rather than to conduct a detailed analysis of the molecular epidemiology and genetic diversity at local levels. To achieve this goal, it was essential to integrate experimental data

with viral genetic information collected at global, regional, and intra-host scales. This integrative approach allowed us to uncover the complex, multifactorial dynamics driving the evolution of these viruses—insights that would not have been possible through localized analyses alone. We fully acknowledge the importance of local epidemiological studies, and additional investigations focused specifically on the molecular epidemiology and implications of GII.17 noroviruses in each location (Germany, Spain, and Argentina) are currently underway.

2. The second approach, a comprehensive analysis of publicly available GII.17 sequences, exhibited significant redundancy. Much of the older strain data reported before 2021, particularly concerning clusters A, B, C, and D (including HBGA binding), has been adequately covered in previous studies. Replicating similar experiments on known clusters and re-analyzing previously published information provides minimal new insight into pre-2021 GII.17 strains. The authors should prioritize the analysis of the current GII.17 strain surge and reorganize their data and discussion to reflect this focus.

Response: We respectfully disagree with these comments. First, including data from earlier viruses (clusters A, B, C, and D), was essential for identifying the traits acquired by the newly emerged GII.17 viruses during their diversification. For example, the pre-2021 GII.17 virus E11161/France/2014 was identified as direct ancestor of the new GII.17 viruses. Our study is the first to demonstrate that this ancestral virus exhibited reduced host-cell binding capacity compared to both with new viruses and previously dominant clusters C and D viruses. Notably, a single mutation in the new viruses resulted in improved host-cell binding, a feature that may have contributed to their predominance in Europe and the America. Second, to our knowledge, no previous study has comprehensively analyzed the antigenic relationship among GII.17 noroviruses. Prior studies only included a small subset of (≤ 4) VLPs without including representative viruses from all four initial clusters (Yi et al. *Emerg Microbes Infect* 2021 DOI: 10.1080/22221751.2021.1925162; Lindesmith et al. *J Infect Dis* 2017 DOI: 10.1093/infdis/jix385; Zhang et al. *Sci Rep* 2015 DOI: 10.1038/srep17687; Strother et al. *Front Immunol* 2023 DOI: 10.3389/fimmu.2023.1229724; Estienney et al. *Front Microbiol* 2022 DOI: 10.3389/fmicb.2022.858245). We included a total of 16 VLPs and 7 groups of mouse sera to demonstrate their antigenic relationship, which successfully provided valuable information on the associations between genetic and antigenic data. Notably, by including viruses from A and B, regarded as not predominant, we delineated residues that could be responsible for the antigenic differences between “endemic and low circulating” (A/B) viruses versus “emerging” (C/D/New) viruses. Third, using this panel of 16 VLPs we also provide confirmatory information on the breadth of binding to saliva samples representing different individuals, eliminating the potential of biases due to the use of unrepresentative viruses from each of those clusters. Taken together, we strongly believe that this study provides a strong overview on the different aspects that lead to the diversification and predominance of the different viruses. Additionally, the antigenic data presented provides unique information for those companies that are currently developing new vaccines and/or are testing vaccine-candidates in clinical trials.

3. The manuscript contains excessive detail and redundant information (both textual and illustrative), which hinders readability. Condensing the text to focus solely on the study's key findings and significant contributions is recommended. Details already established in prior research should be removed, while other less crucial information could be moved to supplementary materials.

Response: We omitted several aspects related previously studies, including part of the original Figure 1. We try to do this without hindering the readability of this study by colleagues working across different viruses and areas of virology (see additional comments through our specific responses).

4. Title needs improvement for clarity: GII.17 noroviruses re-emerged where (Germany, Spain and Argentina? OR worldwide?) and when? (2021-2024?).

Response: Thank you for your suggestion. We have modified the title accordingly.

5. The authors consistently refer to GII.17 phylogenetic clusters as "variants" throughout the manuscript. However, a formal classification system for GII.17 variants has yet to be established. This paper's objective is not to propose an updated classification. It is understood that the ad hoc international norovirus nomenclature working group (Kroneman et al., 2013 and Chhabra et al., 2019), is responsible for assigning names and numbers to norovirus genotypes and variants. It is crucial to determine whether the presented data has been analyzed, discussed, or approved by this group. Also, as previous norovirus classification updates suggest, the naming of variant types should ideally be based on the geographic location of the first identified strains, such as GII.4 New Orleans or GII.4 Sydney. The current use of names like A, B, C, and D warrants clarification as to whether these have been officially approved by the working group. Authors are advised to consider using the term "clusters of interest" in the absence of a properly approved classification system. This approach maintains scientific rigor while awaiting formal guidelines from the designated nomenclature body.

Response: Point well taken. We used “cluster” instead of “variant” throughout the manuscript.

6. In the introduction section, references [15–21] describe GII.17 variants as the Kawasaki-308 cluster, Kawasaki-like, or Kawasaki-2014 like cluster, not using the A-D and "new" designations. To ensure clarity for readers, authors should either utilize publications that employ the A-D nomenclature for GII.17 clusters (e.g., PMID: 26196235; 40476850), or explicitly link the A-D clusters to their commonly recognized names. The 'new' cluster (2021-2014) is most commonly referred to as the Romania-2021 like cluster in recent literature (e.g., PMID: 39328162, 40576844). Authors are advised to align with recent reports to prevent confusion and facilitate the tracking of these viruses.

Response: Thank you for your suggestion. We added references (PMIDs 26196235 and 40476850) according to this reviewer’s suggestion (Page 4 Line 104). Whether clusters A–D and/or the newly identified cluster should be designated as distinct variants remains a topic for discussion by the ad hoc International Norovirus Nomenclature Working Group. In the meantime, we have adopted the A–D cluster naming convention as it aligns with our previous studies (Parra et al. *Emerg Infect Dis* 2015 DOI: 10.3201/eid2108.150652; Parra et al. *PLoS Pathog* 2017 DOI: 10.1371/journal.ppat.1006136), which are now recognized by different research groups (e.g. Chen et al. *Virus Genes* 2020 DOI: 10.1007/s11262-020-01744-6; Liao et al. *Gut Pathog* 2022 DOI: 10.1186/s13099-022-00504-1, Sang and Yang *PeerJ* 2018 DOI: 10.7717/peerj.4333, Lu et al. *J Infect Dis* 2016 DOI: 10.1093/infdis/jiw208), including colleagues of the NoroNet which are also part of the ad hoc international norovirus nomenclature working group (Izquierdo-Lara et al. *Water Res* 2025 DOI: 10.1016/j.watres.2025.124257).

7. Figure 1 can be omitted, as most of its content is already available in previously published work and can be cited accordingly (e.g., PMID: 40476850). Duplicating this information should be avoided.

8. Lines 163-175: To better understand..... decades (Figure 2c). The VP1 amino acid-based phylogenetic tree clearly illustrates the genetic distance between the clusters. However, it is unclear what additional information, beyond cluster distances and the number of amino acid changes, is conveyed by the two-dimensional (2D) sequence space map. For conciseness, authors are requested to streamline this section, moving detailed explanations to the supplementary materials.

10. Lines 185-192: This is previously known information Authors are advised to remove this material from the results section. This can be a part of discussion.

Response: Thank you for your suggestions and comments. We have now consolidated Figures 1 and 2 from the original version into a revised Figure 1. In this revised Figure 1 we decided to keep part of Panel A and Panel B from the original Figure 1 because *Nature Communications* is a multidisciplinary journal with a broad readership, including colleagues working across different viruses and areas of virology. Therefore, this data offers valuable context by illustrating our current understanding of the epidemiology and diversification of this norovirus genotype. Moreover, this figure also supports the identification and clustering of the new GII.17 sequences presented in this study, placing them in a broader evolutionary framework.

Regarding our analyses of amino acid differences and indel patterns among different genotypes and GII.17 clusters, we believe these figures present unique and valuable insights. To our knowledge, this is the first time that quantifiable data, such as those shown in revised Figure 1d, e, and f, have been provided to describe the patterns of amino acid variation and insertions across four of the most prevalent norovirus genotypes and the various GII.17 clusters. Moreover, the unique pattern of indels observed in two evolutionarily intermediate strains (Tokyo27-3/Japan/1976 and Arg13099/Argentina/2015) in the context of the different clusters is novel information and worth presenting (Panel e). We therefore view this information not as redundant, but as complementary to our main findings, offering novel perspectives that have not been previously reported in the field. We added appropriate citations to acknowledge previous studies. Finally, we partially agree with the reviewer that the inclusion of the 2D sequence map may not add substantial value in its current form. While 2D sequence maps are commonly used to illustrate genetic diversity, especially in studies comparing genetic and phenotypic (e.g., antigenic) variation –as shown for influenza and dengue viruses (Smith et al. *Science* 2010 DOI: 10.1126/science.1097211; Katzelnick et al. *Science* 2015 DOI: 10.1126/science.aac5017)– we acknowledge that this comparison was not fully developed in our manuscript. To avoid potential confusion and improve readability, we have now omitted the 2D sequence maps from the revised version.

9. Lines 176-177: ‘a similar pattern shown by GII.4 pandemic viruses 33, 34.’ This can be discussed but it cannot be a part of results of the study.

Response: To increase the readability, this part was omitted.

11. Lines 207-215: ‘Interestingly, the GII.P17..... (Supplementary Figure 4). This new cluster, positioned between GII.P17 and GII.P3, has been previously studied (PMID: 39328162). That prior research concluded that this cluster (thus far identified solely in the United States) remains within GII.P17.

Response: We acknowledge the study by Chhabra et al. (PMID: 39328162, *Euro Surveill* 2025), in which they classified the polymerase type of the new GII.17 noroviruses as GII.P17 (Page 8 Lines 188-189). Regardless, it is interesting as this is an intermediate cluster that could have resulted from convergent evolution on this protein. This has been partially discussed in the original and revised versions.

12. Supplementary Figure 4, which depicts the sub-lineage through a phylogenetic tree, provides no new information. Authors are advised to either condense this information or remove it from the results section.

Response: We respectfully disagree. Supplementary Figure 4 reconfirmed the phylogenetic relationship of GII.P17 and other related polymerase types at the entire ORF1 level. Previous studies and original Figure 3a (revised Figure 2a) presented trees only using RdRp-encoding sequences. To facilitate understanding the idea of Supplementary Figure 4, we presented the trees generated using entire ORF1 and individual protein-encoding sequences in the revised Supplementary Figure 4.

13. Lines 207-2012: Lack of supporting data. Please reword or remove.

Response: Thank you for your comment. We omitted a word “stepwise” to properly describe the data without any additional interpretation.

14. Recombination analysis: A) Recombination in noroviruses is frequently observed within the ORF1/ORF2 region. GII.4 serves as a prime example, where intra-genotype (variant or cluster level) recombination between ORF1 and ORF2 is common. The established dual genotyping or variant typing system largely negates the need for SimPlot analysis to examine ORF1/ORF2 combinations, especially at the intra-genotype level. Additionally, a variant-level classification for GII.17 ORF1 remains undefined. Therefore, please consider removing the figure and condensing the accompanying text. B) The Oregon112 strain originates from the United States, outside the geographical scope of this study. This strain's ORF1 and ORF2 have already been analyzed in a previous study conducted in the United States (PMID: 39328162). To maintain the study's focus, the genome-wide mutation analysis should prioritize the strains identified in the present investigation.

Response: We respectfully disagree with the comment that the established dual genotyping or variant typing system largely negates the need for similarity plot (SimPlot) analyses. This perspective overlooks the potential for different regions of the genome to evolve independently through recombination—a well-documented and highly prevalent mechanism in single-stranded positive-sense RNA viruses, including noroviruses. SimPlot analyses provide a valuable tool to detect such recombination events, which may not be evident through standard genotyping approaches focused solely on partial genome regions. Moreover, since there is not established nomenclature for ORF1 at the intra-genotype level (Chhabra et al. J Gen Virol 2019 DOI: 10.1099/jgv.0.001318), both similarity plot and phylogenetic analyses are the two main tools to identify these events (Bull et al. Emerg Infect Dis 2005 DOI: 10.3201/eid1107.041273; Eden et al. J Virol 2013 DOI: 10.1128/jvi.03464-12).

Regarding the Oregon7112 virus, as we noted in response to your comment #1, the primary objective of this study is to investigate the evolutionary mechanisms underlying the emergence of GII.17 noroviruses, rather than to characterize local genetic diversity in detail. While the Oregon7112 virus was first reported in a previous study (Chhabra et al. Euro Surveill 2024 PMID: 39328162), that work did not include any detailed genetic analysis of this virus. Notably, upon further review of this information we noted an error related to this virus. Their phylogenetic analysis (Figure 3A from Chhabra et al. Euro Surveill 2024) shows that the VP1 of Oregon7112 (GenBank Accession PQ310125) was closely related to recent cluster B viruses (OR685283/CA-RGDS-1180, PQ304634/SanFrancisco0148, PQ304633/SanFrancisco0147, PQ310459/Sheboygan0502), while both our SimPlot and phylogenetic analyses clearly indicates that Oregon7112 virus presents an ORF2/3 (VP1 and VP2) that is similar to the new GII.17 viruses (see our original Figure 3). Phylogenetic trees using a small reference set is attached below for your reference. Of note, there is no modification reported on their original GenBank record (version PQ310125.1 released in October 2024) and we reached same conclusion with re-downloaded sequence data. Because there is uncertainty on what data is incorrect, the GenBank submission (PQ310125) or Figure 3A from Chhabra et al. Euro Surveill 2024, we decide to remove this information from the manuscript.

Figure: Phylogenetic trees of GII.17 norovirus VP1 sequences. (a) The VP1 tree extracted from Chhabra et al. Euro Surveill 2025, Figure 3A (licensed under CC BY 4.0). The original image was not modified, but important reference strains used in panel b are indicated with arrows. The colors of arrows denote GII.17 clusters; blue for cluster A, green for B, orange for C, magenta for D, and red for the new cluster. Those of interests –Oregon7112, CA-RGDS-1180, CA-RGDS-1164 from United States– are highlighted in purple. Please refer to the original paper for the colors used for the strain names on the tree. (b) The VP1 tree was newly generated using reference sequences of GII.17 clusters and strains of interest from the United States (purple highlighted). The colors of the tree branches denote GII.17 clusters. There is a discordance in the position of PQ310125/Oregon7112 on the two trees. This strain is clustered with OR685283/CA-RGDS-1180(cluster B) in the tree from Chhabra et al. (panel a). However, reanalysis indicates that this strain is clustered with OP689689/CA-RGDS-1164 from the new GII.17 cluster (panel b). The clustering pattern observed in panel b verifies our similarity plot data presented in original Figure 3b. (c) Cophylo plot of ORF1 (NS7) and ORF2 (VP1) confirmed different clustering pattern of Oregon7112 virus in two genome regions.

15. HBGA binding to human sera and the study of host cell attachment factor is a complete study in itself with another research perspective. Authors efforts to combine the study with rest of the analysis resulted in overinterpretation of the correlation between predicted antigenic residues and observed epidemiological patterns as there are significant evidence gaps that have not been addressed experimentally. Critical methodological components, site directed mutagenesis, and reverse genetics confirmation of machine-learning predictions, are notably absent. Although the work is described as a “comprehensive antigenic characterization,” its scope is not matched by experimental breadth or

validation. This section need rewriting. The complete section is overly narrative in tone, diffuse in focus, and lacking the experimental substantiation needed to firmly support its conclusions.

Response: It is unclear what the reviewer is trying to request in this paragraph. First, as far as we know “HBGA carbohydrates do not bind to human sera”. If this reviewer meant to comment on the section of “HBGA blocking” assay, we believe we did not do any “overinterpretation” of the data. The analysis was done using a total of 16 VLPs and 7 groups of mouse sera (n=4), which resulted in the largest experimental dataset, a total of 448 HBGA-blockade EC₅₀ titers, to assess the antigenic relationship among the different GII.17 viruses. Second, while we acknowledged that we did not conduct site-direct mutagenesis on VLPs to confirm our machine learning predictions, the prediction of residues was based on the large-scale experimental data and tested by different statistical methods (%variation explained, mean of squared residuals, and %increase of mean squared error). Some of the predicted residues were experimentally verified for GII.17 norovirus by different research groups (Yi et al. *Emerg Microbes Infect* 2021 DOI: 10.1080/22221751.2021.1925162; Lindesmith et al. *J Infect Dis* 2017 DOI: 10.1093/infdis/jix385). In this revised version, we applied the same algorithms and HBGA blockade titers used for GII.4 noroviruses (Kendra et al., *PNAS*, 2021; DOI: 10.1073/pnas.2015874118) and demonstrated that our machine learning models accurately predict residues previously shown -through experimental studies- to contribute to the antigenic diversity of GII.4 noroviruses (e.g., Lindesmith et al. *J Virol* 2012 DOI: 10.1128/JVI.06200-11; Parra et al. *J Virol* 2012 DOI: 10.1128/JVI.06729-11; Tohma et al. *mBio* 2019 DOI: 10.1128/mBio.02202-19). Thus, we hope that the data presented here provide convincing evidence that this method holds substantial potential for identifying critical residues involved in antigenic diversification of this and other human noroviruses. We now expanded our discussion of the machine learning results and the impact this method can have in future studies. Third, we would like to note that although there have been advances in the development of permissive cell lines (Ettayebi et al. *Science* 2016 DOI: 10.1126/science.aaf5211; Ettayebi et al. *mSphere* 2021 DOI: 10.1128/mSphere.01136-20; Ettayebi et al. *mSphere* 2024 DOI: 10.1128/msphere.00448-24) and a plasmid-based reverse genetics system for human norovirus was previously reported (Katayama et al., *PNAS*, 2014; DOI: 10.1073/pnas.1415096111), this system is not yet robust or widely tractable for the type of experiments suggested. As such, these approaches remain beyond the scope of the current manuscript. Finally, we did not analyze or propose any “correlation between predicted antigenic residues and observed epidemiological patterns.” At the most, we suggested a mutation that increased the HBGA-binding to different individuals, antigenic differences, and low anti-GII.17 antibody titers in humans prior the epidemic of the new GII.17, could all facilitate the emergence of this new GII.17 cluster (Page 18 Lines 407-415). All the data mentioned is supported by laboratory-based assays (original Figures 4 and 5; revised Figures 3 and 4).

16. Minimizing extraneous details, this section should concentrate on delineate the significant new information on recent emerging GII.17 strains. Additionally, prior investigations have thoroughly examined HBGA interactions for GII.17 strains identified before the recent surge. To maintain relevance and impact, the analysis presented here, and the subsequent discussion, should primarily address the current GII.17 surge.

Response: We respectfully disagree with this comment. Previous studies have not thoroughly examined HBGA interactions for the GII.17 viruses identified before the recent surge. None of these studies used more than 4 VLPs and none of them used VLPs representing any of the four clusters, let alone the new cluster. Jin et al. (*J Gen Virol* 2016 DOI: 10.1099/jgv.0.000582) reported binding of three GII.17 VLPs (CS-E1 from cluster A, Kawasaki323 from cluster C, and GZ1 from cluster D) using 181 saliva samples with known ABO blood types. Estienney et al. (*Front Microbiol* 2022 DOI: 10.3389/fmicb.2022.858245) characterized the binding of CS-E1 (cluster A), Kawasaki323 (cluster C), and Kawasaki308 (cluster D) VLPs to multiple saliva with known ABO

blood types. They further investigated the binding of these VLPs to duodenal and colonic histological sections. Qian et al. (J Virol 2018 DOI: 10.1128/jvi.01655-18) used two VLPs, 41651/Guangzhou from cluster D and C142 from cluster A to present their binding to different saliva samples from A/B/O donors with structural data. Zhang et al. (Sci Rep 2015 DOI: 10.1038/srep17687) presented binding of only one virus, DG42 (unclear whether is a virus from cluster C or D), to over 60 saliva samples. In contrast to these previous studies, our large panel of VLPs (16 from different GII.17 clusters) with 9 saliva samples from individuals with different HBGA carbohydrates expressions, and the side-by-side evaluations of their interactions, provided concrete evidence of their carbohydrate-binding patterns. First, we confirmed that the difference was not strain-specific, rather, cluster-specific. Second, this study provides binding data of evolutionarily intermediate viruses, Tokyo27-3/Japan/1976 and Arg13099/Argentina/2015, and three new viruses from the new cluster. Third, we used this information to provide evidence that a mutation from ancestral virus E11161/France/2014 was important to increase the breadth of binding to different saliva samples tested in the new cluster viruses. We acknowledged and cited previous studies in the original and revised manuscripts (e.g., “as suggested in previous studies”, “previously identified...”, see Page 11, Lines 240, 244). Thus, we believe the data provided in every single panel of revised Figure 3 is highly important to understand the diversity and evolution of GII.17 norovirus.

September 29, 2025

Dear Reviewers,

Thank you very much for your comments to our revised manuscript (NCOMMS-25-49440A) submitted for publication in Nature Communications. Please find the specific responses to your comments below.

Reviewer #1

The author has provided a detailed and thorough response to my comments, and I have no further supplementary remarks at this time. The article conducts a comprehensive and in-depth study on the evolutionary mechanism of the epidemic caused by the new norovirus variant, and I agree to its publication.

Response: Thank you for your comments, time, and helpful suggestions to improve our manuscript.

Reviewer #2

Due to lack of data sharing among labs, the conclusions in the manuscript might have been understated. In fact, the current reported novel GII.17 strains have been circulating in mainland China during the 2024/2025 winter season. VP1 amino acid sequence of 100% similarity to the reported novel GII.17 has been isolated and sequenced, and the complete genome sequences will be available in the NCBI database soon. The possible already widespread transmission in the inland of China and the ratio of non-secretors in the population (approximately 20%) warrant further investigation of the role of extra binding ability to HBGAs. The authors have mentioned this in their limitations. I wish more excellent work from your lab.

Response: Thank you for your comments and updates regarding the circulation of the new GII.17 cluster in mainland China. We are also aware that outbreaks of these GII.17 noroviruses have recently been reported in Japan and other Asian countries (personal communication with several colleagues and data from Japan: <https://kansen-levelmap.mhlw.go.jp/Byogentai/Pdf/data64j.pdf>). Preliminary sequence analyses indicated that these outbreaks are related to the new GII.17 viruses (2023-2024 sub-cluster). We look forward to seeing the full data and reports and what they reveal related to the genetic and phenotypic evolution of this genotype. We have added a brief comment on the Japanese strains reported in the final sentence of the Discussion section.

Reviewer #3

The revised manuscript “GII.17 norovirus re-emerged in the 2020s as a result of dynamic and adaptive evolutionary processes in its major capsid protein” by Tohma et al. focusses on the current surge of GII.17 noroviruses in Germany, Spain, and Argentina and understanding the genotypic and phenotypic characteristics of noroviruses obtained in this study. While the major strengths of this study include the genomic data from the recent GII.17 viral outbreaks in Germany, Spain, and Argentina; and the machine learning/ binding study using a large panel of VLPs to account for genomic differences both within and between known GII.17 clusters. I also appreciate the authors' effort to accommodate all three reviewers' comments, which has made this version relatively clear. After carefully reviewing the rebuttal and the updated manuscript, I have the following concerns:

Response: We appreciate reviewer’s thorough review and comments. Our responses to each comment are provided below.

Comments:

1. The study effectively outlines its primary objectives in the reporting summary. However, the authors' attempt to provide a comprehensive overview of the GII.17 story, by including previously published information, weakens the overall impact of this study, particularly in the sections discussing genomic variations.

For instance, the two Results sections, "Substitutions, insertions, and deletions play a role in shaping the diversity of GII.17 norovirus" and "Unique ORF1 and ORF3 genes contribute little to viral replication," read more like a literature review and in their current form do not present any new findings. The authors acknowledge that the information presented has been previously reported or explained in other studies, which is not appropriate for the Results section of an original research manuscript.

The Results section should focus on the novel findings derived from the current study's data, with citations and comparisons reserved for the Discussion section. Specifically, these two sections should prioritize the strains examined in this study (from Germany, Spain, and Argentina) as they represent the new information regarding the recent GII.17 surge. All other information in this section should be presented in relation to these specific strains (identified in this study).

Response: We respectfully disagree with your assessment that our current manuscript does not present new findings. From both epidemiologically and evolutionary perspectives, we report highly important new information in the first two Result sections. Specifically, while previous studies (Dinu et al. Arch Virol 2023; Epifanova et al. Arch Virol 2025; Chhabra et al. Euro Surveill 2024) described the sequences and phylogenetic relationships of new GII.17 noroviruses, they did not identify that the new lineage of GII.17 could be divided into several sub-clusters. Our analyses revealed a stepwise divergence across the 2014-2018, 2021-2023, 2022-2024, and 2023-2024 sub-clusters of VP1. Notably, the most recent sub-cluster predominated in Germany (Figure 5), and multiple other countries including Argentina and Spain during 2024-2025 outbreaks. This sub-cluster has been also detected in Japan (see our response to reviewer #2), further supporting its rapid global dissemination. Because of these analyses we were able to pinpoint mutations that seems critical in the evolution and further predominance of this new cluster (Figures 2, 3, and 5). Second, as we noted in our previous response letter, no studies to date have comprehensively analyzed the genetic sequences of GII.17 noroviruses. Our analyses revealed novel findings, including distinct entropy patterns across clusters, inter-cluster pairwise differences, and insertion/deletion events among more than 1,000 viruses, including putative evolutionary intermediates, some of which were selected for phenotypic analyses (Figures 3 and 4). We also observed intra-cluster insertions and deletions, suggesting that the evolutionary dynamics of this genotype are unique as compared to other norovirus genotypes. Third, while previous studies provided phylogenetic trees of RdRp-coding region of new GII.17 noroviruses, none of them identified the parallel and/or back-and-forth mutational pattern in RdRp and other nonstructural proteins. Our findings suggests that these viruses explore multiple mutational pathways in a trial-and-error manner to optimize fitness. This observation aligns with our final analyses (Figure 5), showing that the most recent sub-cluster predominated in Germany with reduced intra-cluster variation as compared to the 'intermediate' sub-clusters. In summary, our genomic analyses go beyond reporting new sequences from Germany, Spain, and Argentina; with the principal finding demonstrating several different evolutionary pathways that ultimately lead the emergence of this virus that is predominating in several countries during the past two years.

2a. Furthermore, to provide a comprehensive overview of the GII.17 noroviruses, important details about the data generated from this study are missing that make the genomic variation story difficult to follow:

a. Please clarify the composition of (a) the 1,400 GII.17 sequences. Was the breakdown 889 archival (are these all GenBank downloads, and if so, when was the last download performed, or were other sources used?), and (b) the composition of the 511 new sequences from surveillance in Germany, Spain, and Argentina?

Response: We apologize for the confusion regarding the numbers of the viral sequence used in this study.

(a) The term “over 1,400 sequences” refers to a total 1,471 sequences, comprising: 68 near full-length GII.17[P17] sequences newly obtained from the samples collected in Germany, Spain, Argentina; 458 partial P2 sequences newly obtained from GII.17 collected over the past 10 years as part of the national surveillance in Germany; and 945 GII.17 sequences (both complete and partial genomes) that were publicly available in GenBank and retrieved on October 18, 2024. In addition, we retrieved 18 non-GII.17 sequences because their polymerase type was closely related to some GII.17 noroviruses as described in the manuscript. In this revised version, we have included an additional supplementary table (Supplementary Table 2), which lists all archival sequences retrieved from GenBank used in this study.

(b) The 511 sequences described in Figure 5 included newly obtained partial P2 and near full-length genome sequences, as well as previously published GII.17 sequences from the German national surveillance system. In this new version of the manuscript, all sequence numbers have been clarified and described with greater accuracy.

2b. Please specify the sequence type, whether full genome, partial, VP1 only, P2 only, for each of the 1,400 sequences. This detail needs to be added to Supplementary Table 1. A new supplementary table should also list all archival sequences used.

Response: We have revised Supplementary Table 1 by adding a column specifying the sequence type (full genome, near full-length, VP1, or P2) for all sequences obtained in this study (national surveillance in Germany, Spain, and Argentina). In addition, as requested, we have created a new Supplementary Table 2, which lists all archival sequences retrieved from GenBank, including their accession numbers, sequence lengths, and corresponding metadata.

2c. Please state how many strains (both archival and new) were used for the analysis from each specific region (ORF1, ORF2, P2, and ORF3).

Response: Thank you for your suggestion. We have now specified in the Methods section the number of viruses used in the analysis for each specific region of the genome.

2d. Currently, there is a discrepancy in the numbers presented. The Results section (lines 144-146) mentions 1,400 genomes for Figure 1b, but the legend for Figure 1b shows 1,013 sequences. Authors should correct this.

Response: Thank you for pointing this out. We have corrected the numbers (n = 1,013 VP1 sequences) in the Results section.

2e. There were 49 full genome sequences generated in this study. Figure 1b highlights 68 VP1 sequences. Please specify the number of VP1 sequences generated in this study.

Response: As noted above, we generated 68 near full-length GII.17[P17] sequences from samples collected in Germany, Spain, Argentina. These sequences were used to generate the phylogenetic tree of VP1 sequences (Figure 1b). Of these, 49 samples collected in Germany were further analyzed to assess intra-host viral diversity.

2f. Another inconsistency exists with the archival sequence count. If 511 sequences are new, the archival count should be 889 (1400 - 511). However, the legend for Figure 1b states 945 archival sequences. This needs to be explained/corrected.

Response: We appreciate the reviewer's careful attention to these numbers. As noted above, a total of 1,471 sequences were included in our analyses: 511 sequences generated from Germany, 20 sequences generated from Spain and Argentina, and 940 archival sequences obtained from GenBank (945 total GenBank sequences minus 5 previously published German sequences). The figure legend has been updated to provide the numbers more accurately.

3. Figure 1b: Please clarify why a 1500-nt sequence was chosen for the VP1 analysis instead of the complete sequence. Only five strains were incomplete, the reason for not removing them from the analysis or excluding them from the complete VP1 dataset should be explained. If these five strains are critical for the analysis, please justify their importance. Specify which end of the VP1 sequence was incomplete (3' or 5'). Explain in the Discussion section, how the missing sequence data might affect the phylogenetic analyses and how it would potentially influence the identification of new clusters associated with the current surge.

Response: As noted in our previous response to Reviewer #1, this criterion was established to include as many sequences as possible in order to capture a comprehensive view of the evolutionary history of this genotype. Specifically, the five sequences with nearly complete VP1 regions (missing only the 5' end) belong to cluster D and were included to minimize potential dataset bias. In the maximum-likelihood framework, missing sites are treated as null and are not incorporated into the likelihood calculation; we described this treatment of missing positions in the Methods section of the original manuscript. Furthermore, the absence of a small portion of the conserved region in these cluster D viruses does not affect the identification of new clusters. To maintain the clarity and flow of the Discussion, we have explained the rationale of this inclusion in the Methods section.

4. Figure 1b is the main figure of this study. Authors should specify whether amino acid or nucleotide sequences were used to construct the VP1 phylogenetic tree in Figure 1b. If nucleotide sequences were used, the authors are advised to regenerate the tree using amino acid sequences to be consistent with the literature for comparison and alignment with recommended criteria for new cluster identification.

Response: Since the beginning of this submission, we have specified that the VP1 tree presented in Figure 1 was generated using VP1-encoding nucleotide sequences. An additional tree using VP1 amino acid sequences is now included in Supplementary Figure 1 (previous Supplementary Figure 2). Although some minor differences in the branching patterns among clusters were observed -which is expected when two-thirds of the information is lost- we did not identify any major inconsistency in the phylogenetic clustering between trees generated using nucleotide and amino acid sequences.

5. Lines 173-175: Consider moving this line in the discussion.

Response: This section describes the results shown in Figure 1d, and we believe it is most appropriately placed in the Result section. While previous studies have independently reported insertions and deletions in specific strains or genotypes, no quantitative data has been provided on the number of sequences with such insertions and deletions summarized across different genotypes.

6. Lines 193-194 and figure 2a: I don't see any phylogenetic divergence in figure 2a that warrants placement of new GII.17 cluster(s) in a new or any known polymerase type. The phylogenetic tree in figure 2a is similar to as shown in reference 28. Authors are advised to please remove or change the text in line 193-194.

7. Lines 198-204: This is not new information. The sub-clustering of polymerase sequences from recent GII.17 strains has already been described in reference 28 as sub-lineage I and II. To avoid literature confusion, please be consistent with already known information while referring to these sub-lineages.

Response: We are not claiming any novelty on the description of the data itself; indeed, reference #28 is cited on that paragraph. That study also noted distinct clustering within the polymerase of GII.17 viruses, which is consistent with our results. In this section, our intention is to describe the outcomes of analyzing archival and novel sequence data, in order to provide context for the novel description of the mutational patterns in ORF1 and ORF3 (Lines 205 – 224). Given the broad scope of this journal, such context is valuable not only for researchers working on norovirus but also for those studying other fast-evolving RNA viruses.

Furthermore, we do not agree with the use of “sub-lineage I and II” for those GII.P17 detected during 2021-2024 as proposed by Chhabra et al. (reference #28). First, the terminology “sub-lineage I and II” implies that both lineages share a common ancestor, which is not supported by our phylogenetic analysis (Figure 2). At present, there is insufficient evidence to determine whether GII.17 viruses from cluster B presenting polymerase “sub-lineage II” originated from GII.P3 or GII.P17 through recurrent mutations in their polymerase sequences. Second, as this reviewer noted during the first round of review, the nomenclature of within the GII.17 genotype and polymerase type remains a topic of active discussion by the ad hoc International Norovirus Nomenclature Working Group. Finally, as explained in our previous response letter, the phylogenetic trees presented in reference 28 appear to contain an error regarding the clustering of viruses within “sub-lineage II,” creating uncertainty about which part of the dataset is reliable. Taken together, we believe it is more appropriate to explain and discuss viruses within GII.P17 without assigning arbitrary names that ignore their evolutionary and spatio-temporal patterns of circulation.

8. Abstract lines 54-56: This statement is true only for the study regions. European countries experienced the GII.17 surge earlier (2023-24) compared to the Americas (2024-2025) and other geographic regions. Additionally, limited genetic diversity was observed among the GII.17 strains circulating in many of these regions. It is possible that sequences from geographic regions undergoing the 2025-26 GII.17 surge might present a different epidemiological picture. Please re-word these lines to avoid generalizing the findings to a global context.

Response: Thank you for your comment. We are now presenting the entropy analyses using global sequence dataset (Supplementary Figure 11) to support this statement. While sampling strategy are different by countries and there could be larger sampling bias to directly compare the data between cluster D and new cluster viruses detected at the global level, the temporal trend of amino acid entropy at the global level resembles to the one observed in Germany.

9. A key strength of this study is the HBGA blocking assay. The revised sections have been improved and are now much clearer.

Response: Thank you for your comment.

10. Lines 182-183: role of residues 393–396 in cluster D has been explained in earlier study (PMID: 28973354). Please discuss accordingly.

Response: While the study by Lindesmith et al. (J Infect Dis 2017 DOI: 10.1093/infdis/jix385) evaluated the role of residues of 393-396 in HBGA-binding and blockade of three GII.17 VLPs, there was not extensive discussion about the role of insertions/deletions. This study was already cited and their contributions to understanding of GII.17 were discussed. In this paragraph, we

specifically discuss the presence of several insertions, deletions, and mutations in this region, extending beyond what could be considered the consensus sequence of viruses belonging to cluster D and the newly identified cluster. This is a critical consideration for understanding the unique evolutionary dynamics of this genotype.

11. The confirmation of cluster-specific binding by the authors replicates previously published findings for GII.17 viruses (e.g., PMID: 28973354). While this study expands the scope by testing 16 VLPs against 9 different HBGA saliva samples, its main conclusion—that binding is cluster-specific—is identical to earlier reports using only 1–4 VLPs. This significant overlap with existing literature must be discussed more robustly.

Response: Thank you for your suggestion. We have provided additional discussion in Lines 470–472 of the Discussion section to further acknowledge previous findings and to reinforce our discussion of the HBGA binding topic (Lines 478–480).